# A network of coiled-coil and actin-like proteins controls the cellular organization of magnetosome organelles in deep-branching magnetotactic bacteria

Virginia V. Russell [1], Anthony T. Iavarone[2], Ertan Ozyamak[1,3], Carly Grant[1] & Arash Komeili [1] ✉

Magnetotactic Bacteria (MTB) are a diverse group of microorganisms that use magnetosomes, organelles composed of magnetite or greigite, to navigate along geomagnetic fields. While MTB span several phyla and exhibit diverse phenotypes, magnetosome formation has been mechanistically studied in only two species of *Alphaproteobacteria*. Here, we use *Desulfovibrio magneticus* RS-1 to uncover the mechanisms behind tooth-shaped magnetosome assembly in deep-branching MTB. Our findings show that magnetic particles in RS-1 initially form randomly within the cell before localizing to the positive cell curvature. Genetic and proteomic analyses indicate that early biomineralization involves membrane-associated proteins found in all MTB, while later stages depend on coiled-coil (Mad20, 23, 25, and 26) and actin-like (MamK and Mad28) proteins, most of which are unique to deep-branching MTB. These findings suggest that while biomineralization originates from a common ancestor, magnetosome chain organization has distinct evolutionarily origins among different MTB lineages.

Magnetotactic Bacteria (MTB) are a diverse group of microorganisms that navigate along geomagnetic fields to locate low-oxygen environments in a process termed magneto-aerotaxis. This ability relies on magnetosomes—specialized organelles containing magnetic crystals of magnetite ($Fe_3O_4$) or greigite ($Fe_3S_4$) 50-70 nm in diameter, enclosed by a lipid bilayer membrane with a unique set of proteins[1,2]. Depending on the species, magnetosomes form single or multiple chains, that allow for alignment with external fields[3]. These features have made MTB models for studying bacterial organelle formation, biomineralization, and the global geochemical cycling of iron[4]. MTB have also been deployed as potential vehicles for a variety of biotechnological applications[5]. A key bottleneck in the study of MTB has been the limited number of model organisms that sufficiently represent the mechanistic diversity and evolution of magnetosome formation.

MTB are a polyphyletic group of microorganisms that belong to *Alphaproteobacteria*, *Gammaproteobacteria*, *Candidatus Etaproteobacteria* classes in the *Pseudomonadota* phylum, and the phyla *Desulfobacterota*, *Nitrospirota*, *Candidatus Omnitrophota*, and *Elusimicrobiota* (Fig. 1B). Metagenomic studies suggest MTB may also exist in the *Latescibacteria*, *Planctomycetes*, *Nitrospinota*, *Fibrobacterota*, *Riflebacteria*, *Hydrogendentota*, *Myxococcota*, *Bdellovibrionota* phyla and possibly more[2,6]. MTB exhibit diverse phenotypes, including variations in crystal shape, chain organization, and magnetosome composition[3]. Phylogeny correlates with some of these features; for instance, *Alphaproteobacteria* produce cubooctahedral and prismatic magnetite crystals, while *Desulfobacterota* and other deep-branching MTB (*Nitrospirota*, *Omnitrophota*, *Elusimicrobiota*) form irregular tooth-shaped magnetite

[1]Plant and Microbiology, University of California Berkeley, Berkeley, CA, USA. [2]QB3/Chemistry Mass Spectrometry Facility, University of California Berkeley, Berkeley, CA, USA. [3]Present address: Bio-Rad Laboratories, Hercules, CA, USA. ✉e-mail: komeili@berkeley.edu

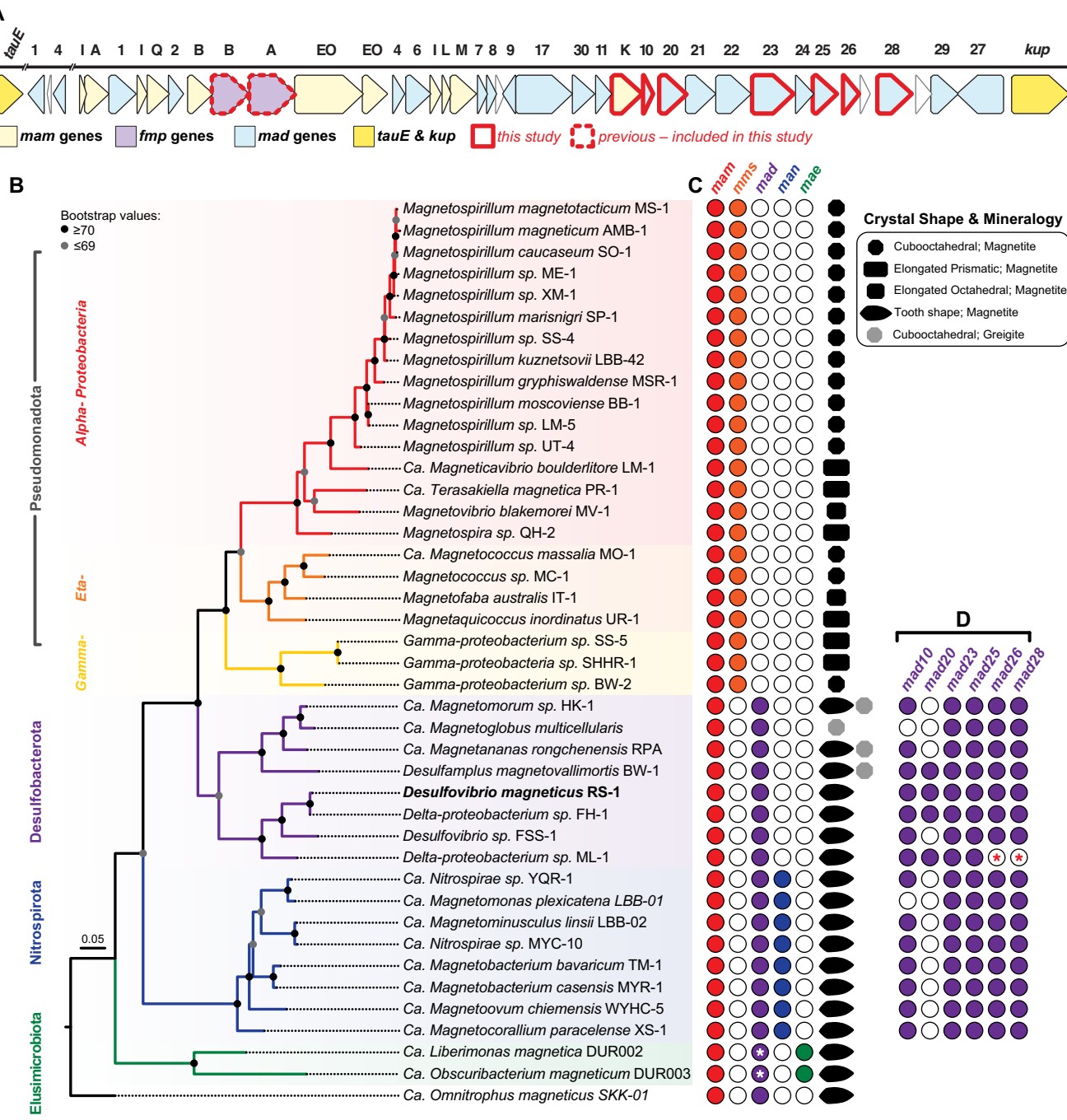

**Fig. 1 | Phylogenetic tree of MTB with their magnetosome gene composition and magnetosome crystal shape. A** Magnetosome gene cluster in RS-1. All *mam* genes are in yellow, all *mad* genes are in blue, and *fmp* genes are in purple. Genes outlined in red (*mamK, mad10, mad20, mad23, mad25, mad26* and *mad28*) were the focus of this study. Outlined in dotted red (*fmpA* and *fmpB*) are genes mutated in a previous study[12], but examined further in this study. **B** 16S maximum likelihood tree. MAFFT was used for alignment of 16S genes. Maximum likelihood tree was built using PhyML Tree[52] with GTR substitution model and 1000 bootstraps. **C** For each MTB species the classification of the type of magnetosome genes (*mam, mms,*

*mad, man,* or *mae*) are provided as well as the magnetosome mineral composition and shape. White asterisks indicate the potential presence of *mad* genes, which were previously reported as missing[8] (*see* Supplementary Fig. 1B). **D** Deep-branching MTB species that contain the *mad* genes studied here. Red asterisks represent uncertainty due to a lack of a completed genome on NCBI. Supplemental Table 1 contains all the data (NCBI genome IDs, 16S gene ID, and references) used to generate the phylogenetic tree **B**, magnetosome gene information **A**, **C**, & **D**, and literature with crystal TEM images and chemical composition **C**.

crystals thought to represent the ancestral magnetosome form (Fig. 1C)[3].

Despite the vast diversity of MTB, most mechanistic studies have focused on two Alphaproteobacterial species, *Magnetospirillum magneticum* AMB-1 and *Magnetospirillum gryphiswaldense* MSR-1, which contain a conserved magnetosome gene cluster (MGC). The MGC

spans ~100 kilobases and includes 50–100 genes, with 29 necessary and sufficient for magnetosome formation[7]. Depending on the accounting, only 5–9 core genes (*mamA, B, E,* I, K, M, O, P and *Q*) are shared across all MTB[8,9]. However, many genes critical for magnetosome formation in *Alphaproteobacteria* are absent in other MTB. In turn, group-specific genes are found in the MGCs of other MTB. For

instance, *mad* genes, originally named for '**m**agnetosome-**a**ssociated ***D**eltaproteobacteria*,' are present in all deep-branching MTB. Similarly, *man* (**m**agnetosome-**a**ssociated **N**itrospirota) genes are found in *Nitrospirota*, and *mae* (**ma**gnetosome **E**lusimicrobiota) genes are found in *Elusimicrobiota*[8]. Since few non-Alphaproteobacterial MTB are cultured, and most lack genetic systems, mechanisms of *mad* gene function and irregular tooth-shaped magnetosome formation have remained elusive. This is a critical barrier in the study of MTB as tooth-shaped magnetosomes are common in nature[3], yet their biomineralization mechanisms remain unknown.

The most promising model for deep-branching MTB is *Desulfovibrio magneticus* RS-1, a sulfate-reducing obligate anaerobe from the *Desulfobacterota* phylum (originally grouped within *Deltaproteobacteria*)[10,11]. RS-1 synthesizes a single chain of irregular tooth-shaped magnetite magnetosomes, organized into subchains of contiguous crystals separated by gaps along the positive cell curvature (Fig. 2C-ii and G). It is one of the few cultured non-Alphaproteobacterial MTB with a sequenced genome and the only deep-branching strain with tools for heterologous gene expression and genome editing[12–14]. Previous studies of RS-1 have highlighted several features distinguishing it from Alphaproteobacterial models. First, RS-1 produces tooth-shaped crystals. Second, unlike *Alphaproteobacteria*, mature RS-1 crystals appear to lack a lipid membrane[15], though its MGC contains proteins with predicted transmembrane domains, many of which are homologs of well-studied Mam proteins. Genetic studies show mutations in these genes disrupt magnetite formation, implying the involvement of a magnetosome membrane at some stage[12]. Finally, RS-1, like all deep-branching MTB, contains a suite of *mad* genes. One conspicuous feature of many Mad proteins is that they are predicted to contain one or multiple coiled-coil segments. These domains typically mediate homo- and hetero-polymeric interactions allowing for assembly of simple dimers to complex multimeric structures. In bacteria, proteins with coiled-coil domains are involved in organization and segregation of chromosomes, assembly and function of the type III secretion system, and formation of intermediate-like filaments that influence cell shape[16–19]. Bioinformatic analyses of deep-branching MTB have proposed roles in biomineralization, protein sorting, and chain organization for coiled-coil-domain containing Mad proteins[9]. However, direct experimental evidence of their function has remained mysterious.

Here, we comprehensively characterize the RS-1 biomineralization process to develop a deeper understanding of the mechanisms and evolution of magnetosome formation in deep-branching MTB. Using a conditional biomineralization assay combined with genetics and proteomics, we define specific steps in magnetic particle growth and identify specific factors associated with the progression of biomineralization and magnetosome chain formation. We find that early steps of biomineralization utilize proteins enriched in transmembrane domains including some encoded by *mam* genes common to all MTB. In contrast, localization, organization, and positioning of the magnetosome chain rely on the suite of coiled-coil domain-containing Mad proteins as well as the actin-like proteins, MamK and Mad28. These findings indicate a complex evolutionary history where iron biomineralization is an ancestral feature of all MTB while chain formation follows an independent and group-specific path.

## Results

### Hydrogen inhibition of magnetosome synthesis

Our first goal was to develop a robust system to visualize the steps of magnetosome formation in RS-1. In other MTB, magnetite is not formed under iron-limiting conditions. Addition of iron then triggers synchronized magnetosome chain development[20]. However, transitioning from iron-limited to iron-replete conditions in RS-1 leads to the accumulation of ferrosomes[15]—organelles involved in storage and detoxification of iron[21,22]. The abundance and appearance of ferrosomes in electron microscopy obscure magnetosomes, making it impossible to visualize the early stages of magnetite biomineralization. To overcome this limitation, we sought an alternative method of magnetosome inhibition that avoids iron starvation.

Serendipitously, we discovered that growth in hydrogen instead of nitrogen headspace significantly reduces RS-1's magnetic response (measured as the coefficient of magnetism ($C_{mag}$)[23]) (Fig. 2A). A trace $C_{mag}$ at high hydrogen concentrations suggested the need for hydrogen exposure throughout the culture medium. To test this, we flushed both the headspace and medium with 5% hydrogen and grew cells while spinning. Under these conditions, cultures showed no magnetic response as indicated by a $C_{mag}$ of 1 (Fig. 2B), and absence of magnetosomes when viewed by transmission electron microscopy (TEM) (Fig. 2C).

One explanation for these observations is that hydrogen inhibits RS-1 growth, indirectly preventing magnetosome formation. However, growth measurements showed similar final cell densities in hydrogen and nitrogen (Fig. 2D), with hydrogen-grown cells exhibiting faster doubling times (6 vs. 8 hours) (Fig. 2D). Another possibility is that hydrogen limits iron uptake, restricting magnetosome production. To test this, we used the post-iron-starvation accumulation of ferrosomes as a proxy for iron uptake. Cells were grown in hydrogen and iron-limiting conditions and subsequently exposed to iron with either a nitrogen or hydrogen headspace. Ferrosomes were formed in both nitrogen and hydrogen (Fig. 2E iii and iv), but magnetosomes formed only in nitrogen conditions (Fig. 2E-iv). This indicates hydrogen does not inhibit iron uptake. Finally, we showed that hydrogen growth does not have a significant impact on the global expression of magnetosome proteins as compared to nitrogen growth conditions. Thus, the mechanisms for inhibition of biomineralization under hydrogen growth conditions remain unresolved (See supplementary notes).

### Characteristics of early and late stages of biomineralization

We next hypothesized that switching from hydrogen to nitrogen conditions under iron replete conditions could restore magnetosome formation without ferrosome interference. To test this, RS-1 was grown with 10% hydrogen to eliminate magnetosomes, then transferred to nitrogen infused medium. As predicted, magnetic response increased over time (Fig. 2F), no ferrosomes were formed, and only magnetosomes were observed (Fig. 2Gi-iii). This transition provides a reliable method to synchronize magnetosome formation and study biomineralization.

To capture early stages of biomineralization, RS-1 was incubated shaking in hydrogen-infused medium for several passages to eliminate existing magnetosomes (Fig. 2B-i). Cells were then transferred to nitrogen-infused medium and harvested once a measurable magnetic response ($C_{mag}$ 1.05−1.15) was detected. Early timepoints in biomineralization showed an average of 3 magnetosomes per cell (Fig. 3A, C). Many cells contained both mature and immature crystals (Fig. 3A-ii and iv). The median crystal length was 38 nm, with a bimodal distribution indicating both immature and mature crystals present (Fig. 3D). Similarly, the shape factor (width-to-length ratio) followed a bimodal pattern, with peaks around 1 for immature and 2 for mature crystals (Fig. 3E). During the late stages of biomineralization ($C_{mag}$ 1.3−1.5), cells contained complete magnetosome chains composed of smaller subchains (Fig. 3B), averaging 12 crystals per cell (Fig. 3C), all localized to the positive cell curvature. The average crystal length was 51 nm with a shape factor of 1.7, both fitting a normal distribution due to presence of fewer immature crystals (Fig. 3D, 3E). These findings point to a crystal growth pattern similar to that proposed for other deep-branching MTB[24]. RS-1 crystals nucleate and grow equidimensionally to approximately 30 nm before elongating anisotropically at one end to produce a tooth-shaped morphology (Fig. 3F). High-magnification TEM images revealed two mature magnetosome morphologies: curved and straight which may indicate divergent paths of crystal

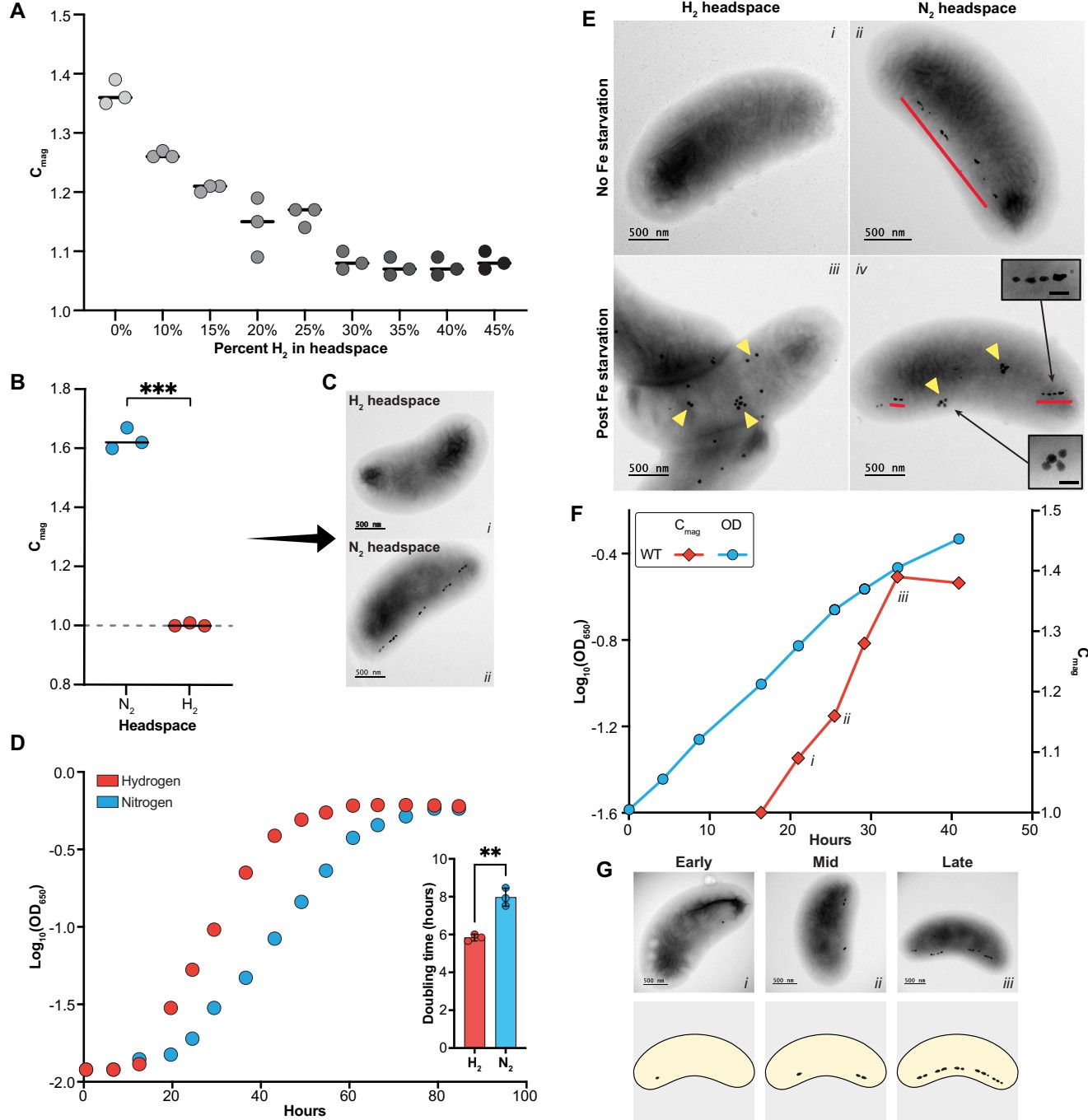

**Fig. 2 | Hydrogen inhibits magnetosome synthesis but not ferrosome production. A** Biological triplicate cultures were grown with varying concentrations of hydrogen gas added to the headspace. Magnetic response ($C_{mag}$) was tracked during growth. Lines with individual points show the median $C_{mag}$ for each hydrogen level. **B** RS-1 cultures were grown in triplicate while rotating on a wheel, with either nitrogen (blue) or hydrogen (orange) in the headspace. $C_{mag}$ measurements showed no magnetic response under hydrogen. A two-tailed Welch's t-test comparing $C_{mag}$ values between nitrogen- and hydrogen-grown cultures yielded a p-value of 0.0009 (Supplemental Table 5). **C** TEM images of the cultures from **B**. **D** Growth curves of wild-type RS-1 grown with nitrogen (blue) or hydrogen (orange) in the headspace. The inset shows mean doubling times; bars represent the mean, and error bars indicate the standard deviation from three biological replicates. A two-tailed Welch's t-test yielded a p-value of 0.0093 (Supplemental Table 5). **E** Initial cultures were either grown with or without iron to simulate iron starvation. Pre-cultures were transferred into either hydrogen or nitrogen

conditions. **E** *i* TEM image of a cell grown in hydrogen with no iron starvation, **E** *ii* TEM image of a cell grown in nitrogen with no iron starvation, and (**E** *iii* and *iv*) are TEM images 40 hours after iron starvation for cells grown in hydrogen (*iii*) or nitrogen (*iv*). Red lines indicate magnetosomes and yellow triangles point to ferrosomes. All cultures in (**E**) were grown in triplicate, and around 100 TEM images were examined for each condition. **E** *i–iv* show the most common phenotype observed under each condition. **F** Growth curve (blue line) of a culture transitioning from hydrogen to nitrogen conditions to initiate biomineralization. The $C_{mag}$ curve (orange line) of the same culture demonstrates biomineralization initiation and progression during growth. **F** *i-iii* indicate time points at which samples were collected for TEM imaging in **G** *i-iii*. **F**, **G** Biomineralization time course of WT was cultured in triplicate. Around 100 images were captured across different biomineralization stages—early, mid, and late—as shown in **G**. **G** represents the most common phenotype visualized at each stage.

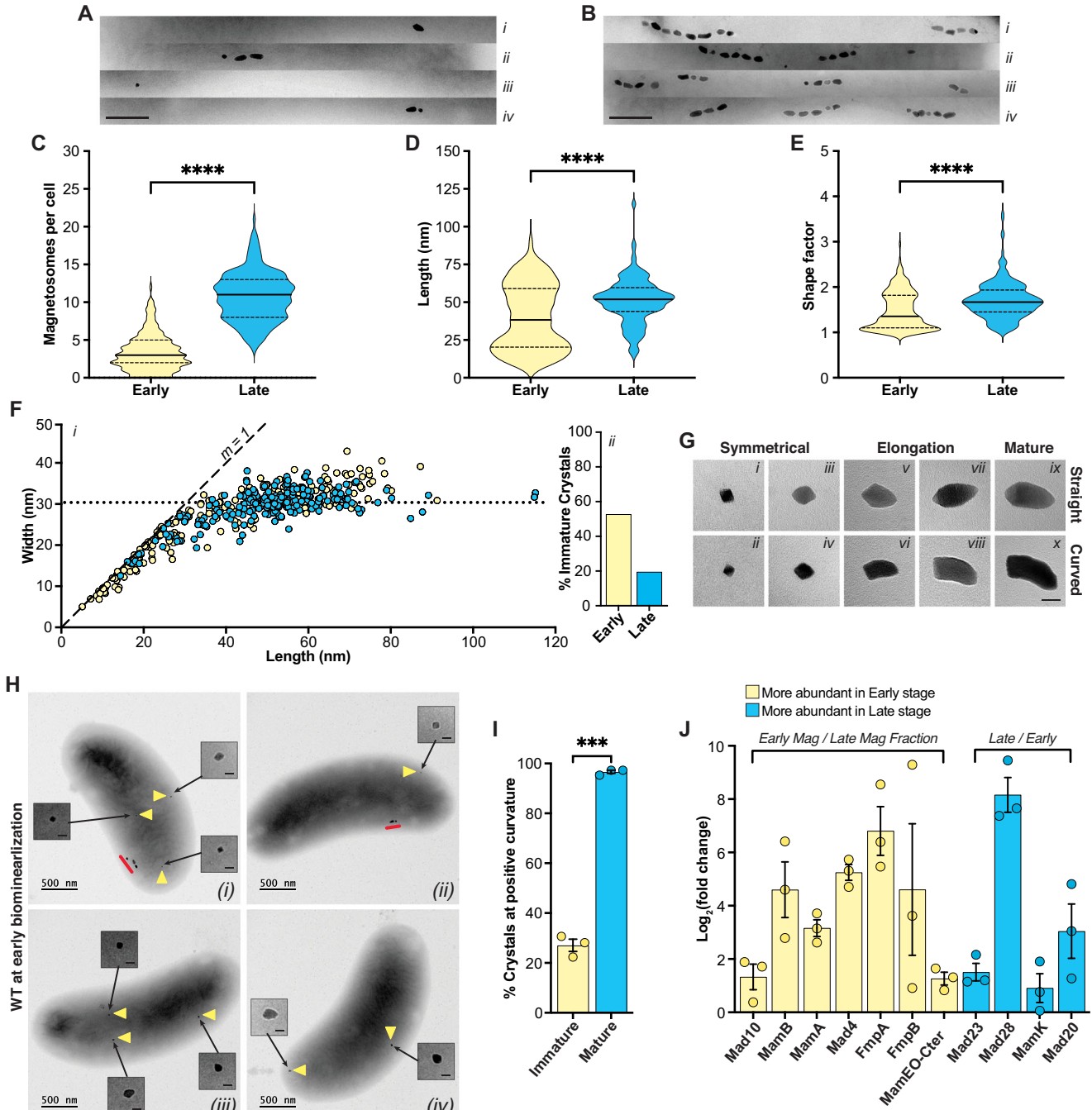

**Fig. 3 | Proteomic analysis of magnetosome proteins at different stages of biomineralization. A, B** TEM images of chains representing early and late stages of biomineralization, with the scale bars representing 250 nm. **C** Violin plots depicting the number of magnetosomes per cell for cells in early and late stages of biomineralization. **D** Violin plots illustrating measurements of magnetosome length and **E** shape factor (length/width) for crystals found in early and late stages of biomineralization, $n = 212$ for early stages and $n = 214$ for late stages. Two-tailed Mann-Whitney U tests yielded p-values of <0.0001 for **C**–**E** (Supplemental table 6). **F** Each point is the measurement of the length and width of one crystal from WT (AK80). Yellow dots represent crystals from cells at early stages of biomineralization. Blue dots represent crystals from cells at late stages of biomineralization. The dotted line represents the median width of crystals from late stages of biomineralization. Dashed line shows a slope (m) of 1. **G** TEM images of RS-1 magnetosomes at various stages of crystal growth. Panels (*i*–*x*) illustrate different stages of crystal growth observed and identified from analysis of over 400 TEM images and measurements

of more than 400 individual crystals, as quantified in **D**, **F**. The black scale bar in panel (*x*), representing 25 nm, applies to all panels (*i*–*x*). **H** TEM images of early stage biomineralization of WT cells. Yellow triangles point to immature crystals that are included as zoomed-in images in the inset. Red lines indicates mature magnetosomes. The scale bars in the insets represents 25 nm and applies to panels *i* through *iv*. **I** The mean proportion of magnetosomes positioned at the positive curvature of the cell for both immature (yellow bar) and mature (blue bar) crystals during this stage. Three biological replicates with a total of 140 cells were counted for each condition; $n = 254$ immature crystals, and $n = 394$ mature crystals and error bars represent±SEM. A two-tailed Welch's T-Test yielded a p-value of 0.0006 for **I** (Supplemental table 6). **J** The mean log$_2$ maximum fold change of magnetosome protein abundance detected in the magnetosome fraction at different stages of biomineralization for three biological replicates. The error bars represent±SEM. Yellow bars represent proteins more abundant at the early stage and blue bars represent proteins more abundant at the late stage of biomineralization.

growth during the elongation step (Fig. 3G v to 3G x). However, additional analysis is required to confirm whether these are true characteristics of the 3D crystals, since the images in this study capture only 2D projections.

While observing biomineralization stages, we discovered that subcellular magnetosome location correlates with crystal maturation. In early stages, nearly all mature crystals are positioned at the positive cell curvature, where magnetosome chains typically form, but only about 25% of immature crystals are located there (Fig. 3H, I).

## Distinct proteomes differentiate early- and late-stage magnetosomes

We next investigated if specific cohorts of proteins are associated with distinct biomineralization stages. Cells from early and late stages were harvested and lysed, and magnetosomes were isolated using a magnetic column. Proteomic analysis of cell lysates and magnetosome fractions via liquid chromatography-mass spectrometry focused on RS-1 MGC-encoded proteins enriched at each stage. Early-stage magnetosomes were enriched for Mad10, MamB, MamA, Mad4, FmpA, FmpB, and MamEO-Cter (Fig. 3J), many of which contain predicted transmembrane domains (Supplementary Fig. 1A). Late-stage magnetosomes were enriched for Mad20, Mad23, Mad28, and MamK, all of which lack transmembrane domains (Supplementary Fig. 1A). The actin-like proteins, Mad28 and MamK (Supplementary Fig. 1A), may have roles in chain alignment and organization based on previous work in Alphaproteobacterial MTB[25]. These results suggest a shift in membrane association and chain organization from early to late stages.

## Early magnetosome proteins regulate biomineralization activity

We next turned to genetics to provide functional evidence for stage-specific action of magnetosome proteins. Previous studies using chemical and UV mutagenesis identified RS-1 mutants with severe biomineralization defects, including several alleles of *mamB* that lack a magnetic response and do not produce crystals[12]. This phenotype corroborates our finding that MamB preferentially associates with early-stage magnetosomes (Fig. 3J).

In contrast to *mamB*, mutants in *fmpA* and *fmpB* can still produce some magnetic particles[12]. FmpA has a transmembrane domain, whereas FmpB contains a signal peptide. Both proteins share serine protease and denitrogenase Fe-Mo domains and are enriched on magnetosomes in the early stages of biomineralization (Supplementary Figs. 1A and Fig. 3J). Interestingly, the mutations in both *fmpA* and *fmpB* are predicted to eliminate their respective denitrogenase Fe-Mo domains (Fig. 4 and Supplementary Fig. 1A). Detailed TEM analysis revealed distinct defects for each strain: *fmpA* mutants produce misshaped circular or symmetrical crystals (Fig. 4A, G and Supplementary Fig. 2-A), while *fmpB* mutants form smaller wildtype-like (WT-like) crystals, some of which are misshaped (Fig. 4B, H and Supplementary Fig. 2-B). Consistent with previous findings[12], both mutants exhibit significantly fewer crystals per cell, smaller crystal sizes, and a reduced shape factors as compared to WT (Fig. 4D, 4E, F).

In addition to the defects in crystal shape and size, *fmpA* and *fmpB* mutants exhibited abnormal crystal placement with most crystals not localized to the cell's positive curvature (Fig. 4J). The crystals of the *fmpA* mutant also fail to form chains, while only 5% of *fmpB* mutant cells show chain organization (Fig. 4I). These findings suggest FmpA and FmpB are crucial for early magnetosome synthesis and indicate that biomineralization likely initiates away from the positive curvature. We propose that crystal nucleation occurs throughout the cell and crystals are later transported to the positive curvature during the maturation process. Without FmpA and FmpB, crystals could remain "stuck" at nucleation sites, blocking their relocation and preventing new crystal formation, resulting in fewer crystals in mutant cells (Fig. 4D).

## Initiation of chain formation by Mad10

Previous studies identified Mad10 as a magnetite-binding protein[26,27] and hypothesized a role in magnetite nucleation or shape control. We find that Mad10 has the highest abundance ratio among MGC proteins at the magnetosome compared to the cell lysate under all conditions (Supplementary Fig. 3A-C). To investigate its function, we deleted *mad10* in RS-1 via allelic exchange. The Δ*mad10* mutant shows a significantly reduced magnetic response, which is restored by plasmid-based *mad10* expression (Supplementary Fig. 4B and C). Despite the low $C_{mag}$, however, crystal size and shape remain unchanged and, instead, magnetosomes cluster in a single location within the cell (Fig. 5A, Supplementary Fig. 4I and Supplementary Fig. 5A). Crystal numbers also vary widely per cell (Supplementary Fig. 4D), likely due to unequal distribution during cell division. These findings indicate that Mad10 is essential for magnetosome chain organization but not for crystal size or shape regulation (Fig. 5A).

## A module of Mad proteins assembles the magnetosome subchain

We next targeted genes encoding proteins more abundant in late stages of biomineralization (Fig. 3J) and neighboring genes potentially belonging to the same operon (Fig. 1A). All deletions were successfully complemented via plasmid-based expression (Supplementary Fig. 4C, Supplementary Fig. 6A-G). Mutants exhibited a range of chain formation defects which were quantified based on chain placement, magnetosome distribution, and chain/subchain phenotypes (Supplementary Fig. 7 F-I).

This analysis identified Mad20, Mad23, Mad25, and Mad26 as a module for magnetosome subchain assembly in RS-1. The Δ*mad23* mutant produces the same number of magnetosomes per cell as the WT but they appear dispersed throughout the cell (Fig. 5C and Supplementary Fig. 4D). This mutant also has the fewest crystals per subchain among the chain organization mutants and WT resulting in the lowest $C_{mag}$ amongst these strains (Fig. 5C, Supplementary Fig. 4B, Supplementary Fig. 4E, and Supplementary Fig. 5C). These results suggest that Mad23 connects individual magnetosomes to form subchains. In contrast, the Δ*mad20* mutant forms distinct subchains, some linear, and others in the form of rings, clusters, or curved lines (Fig. 5B and Supplementary Fig. 5B). It has fewer subchains than WT (Supplementary Fig. 4F and Supplemental 5 B), but each contains more magnetosomes (Supplementary Fig. 4E). This suggests Mad20 regulates crystal influx into subchains; without it, excess crystals cause curling and distortion (Fig. 5B and Supplementary Fig. 5B). Δ*mad25* and Δ*mad26* mutants show similar phenotypes (Fig. 5D-E and Supplementary Fig. 5D-E), with crystal and subchain numbers comparable to WT (Supplementary Fig. 4D and F). However, most subchains cluster or form rings, especially in Δ*mad26*, where ~80% cluster and 15% form rings (Supplementary Fig. 7E). These findings indicate Mad25 and Mad26 are essential for maintaining subchain linearity.

Overall, these mutants reveal a variety of defects in magnetosome chain assembly and localization in RS-1. Strikingly, many of the mutants exhibit clusters, rings, or curved subchains within the same cell hinting at a lack of long-range connections between these structures. As such, these data suggest that subchains are a fundamental structural unit of chain architecture in RS-1.

## MamK and Mad28 control subchain separation and chain localization

Mad28 and MamK were selected for genetic analysis due to their higher abundance in late-stage magnetosomes (Fig. 3J) and their actin-like domains (Supplementary Fig. 1A and S8). *mamK* is present in nearly all MTB and is well-studied in AMB-1 and MSR-1[28–30]. In AMB-1, *mamK* deletion disrupts chain cohesion[29], creating gaps that affect equal distribution during cell division[31]. In MSR-1, *mamK* deletion results in mislocalized chains, mostly away from the mid-cell[32].

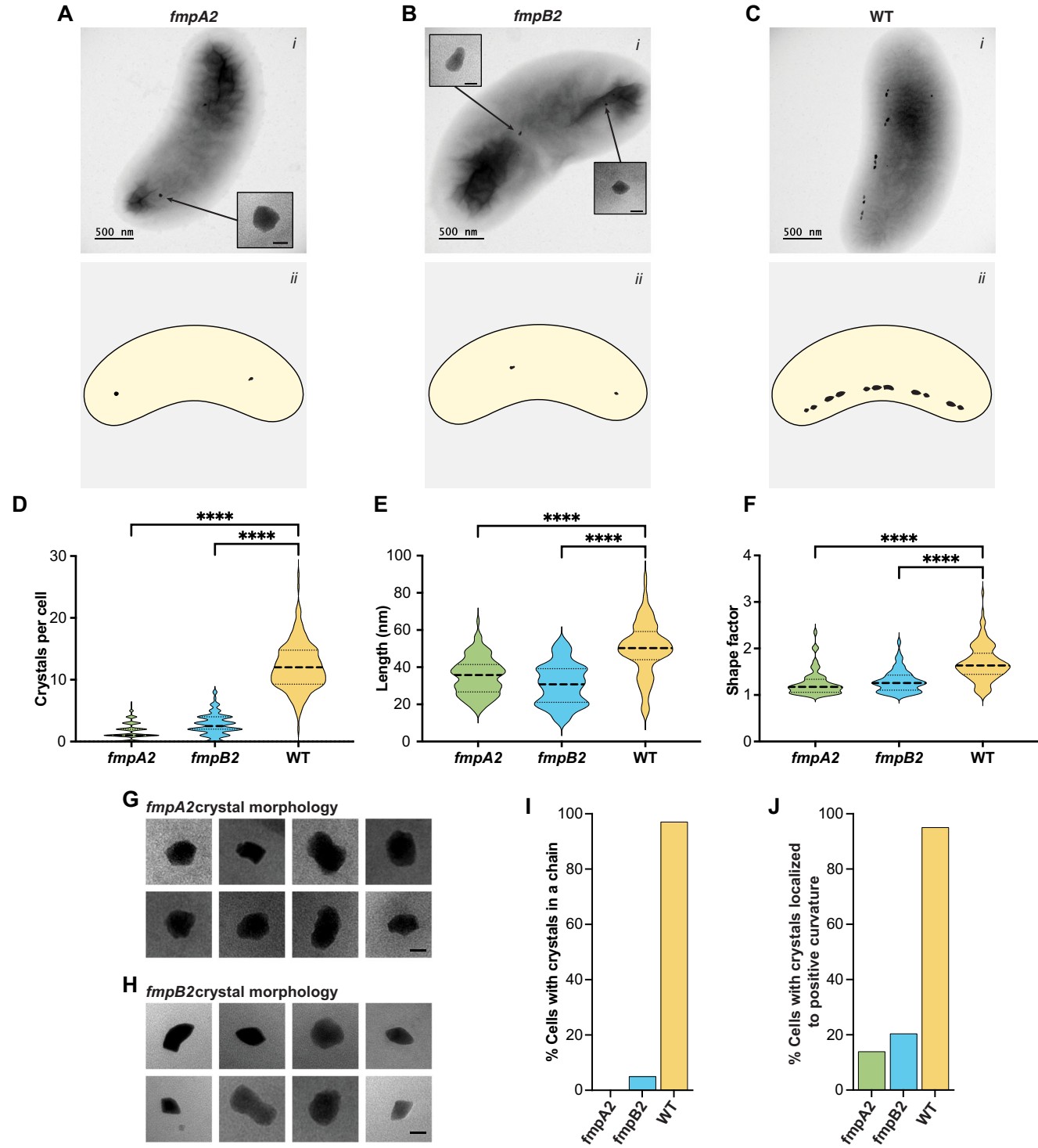

**Fig. 4 | Characterization of *fmpA* and *fmpB* mutants. A**–**C** (*i*) TEM images of *fmpA* and *fmpB* mutant strains (scale bars 500 nm for main image and 25 nm for inset). *fmpA2* represents the *fmpA* allele 2 mutant Q418*, where there is a premature stop at amino acid 418 and *fmpB2* represent the *fmpB* allele 2 mutant 808delG, which has a frameshift from a deleted nucleotide at amino acid 270[12] (Supplementary Fig. 1A). (*ii*) Cartoon diagram of each strain. **D** Violin plots showing the number of crystals per cell for each strain (n = 204). **E** Violin plots showing magnetosome length (n = 114), and **F** shape factor (length/width) for crystals in each strain (n = 114). Two-tailed Mann-Whitney U tests and one-way ANOVA yielded p-values < 0.0001 for all comparisons (Supplemental Table 7). **G**, **H** Crystal morphology for each mutant, scale bars are 25 nm. **I** Percentage of cells exhibiting a chain phenotype, defined as having more than one crystal aligned in a row, for each strain. **J** Percent of cells with crystals located at the positive curvature in the cell. **I**, **J** 200 cells were counted.

Similarly, in RS-1, we find that MamK plays a role in chain organization. Deletion of *mamK* eliminated the subchain phenotype (Fig. 5G, Supplementary Fig. 4F, and Supplementary Fig. 5F), leading to a single continuous chain, which sometimes wrapped around itself, resembling

multiple chains. These mutants also had more crystals per cell compared to WT (Fig. 5N, Supplementary Fig. 4D).

RS-1, and all deep-branching MTB, contain another magnetosome gene, *mad28*, that encodes a protein with an actin-like domain

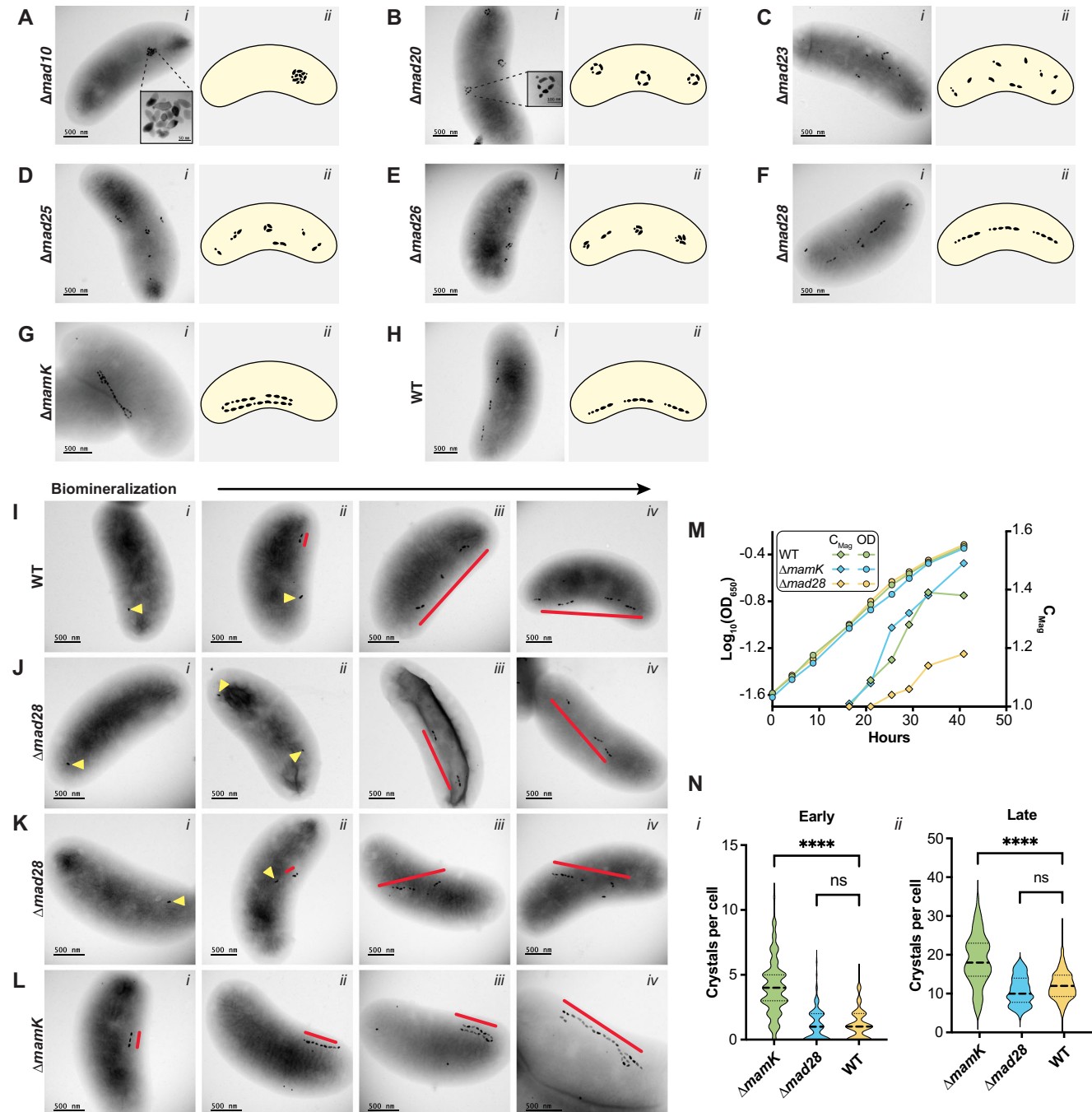

**Fig. 5 | Deletions of *mad* genes and *mamK* result in a variety of chain formation and localization defects. A–H** Phenotypes of different magnetosome gene deletion mutants and WT RS-1 with scale bars representing 500 nm. For each strain, (*i*) shows a representative TEM image and (*ii*) presents a corresponding cartoon illustrating typical magnetosome chain morphology. All strains were grown in triplicate, and over 200 TEM images were analyzed per strain. More than 200 cells per strain were categorized based on chain placement, magnetosome distribution, overall chain phenotype, and subchain characteristics (see Supplementary Figs. 4 and 7). C$_{mag}$ values were also measured from each strain's triplicate cultures (see Supplementary Fig. 4). **I–L** TEM images showing the biomineralization time course of Δ*mad28*, Δ*mamK*, and WT. Scale bars represent 500 nm. Yellow triangles mark individual magnetosomes, and red lines highlight magnetosome chains or subchains. Each strain was cultured in triplicate, and ~100 images were captured across early, mid, and late biomineralization stages. **I, L** display the most common

phenotypes for WT and Δ*mamK*, respectively. For Δ*mad28*, two distinct phenotypes were observed: one with chain formation at the mid-cell **J** and another with diagonal or transverse chain localization **K**. Δ*mamK* **L** showed no formation of subchains. **M** Growth curve and C$_{mag}$ curve for WT, Δ*mad28* and Δ*mamK*. **N** *i* Violin plots showing the number of magnetosomes per cell for WT, Δ*mad28*, and Δ*mamK* during early stages of biomineralization. **N** *ii* Violin plots showing the number of magnetosomes per cell for the same strains during late stages of biomineralization. Over 100 cells were counted per strain (Δ*mad28* early *n* = 102, late *n* = 110; Δ*mamK* early *n* = 104, late *n* = 173; WT early *n* = 104, late *n* = 200). Two-tailed Mann-Whitney U tests and one-way ANOVA comparing Δ*mamK* to WT yielded p-values < 0.0001 for both N (*i*) and (*ii*) (Supplemental Table 8). The same tests comparing Δ*mad28* to WT showed no significant differences in the number of magnetosomes per cell at either stage (Supplemental Table 8).

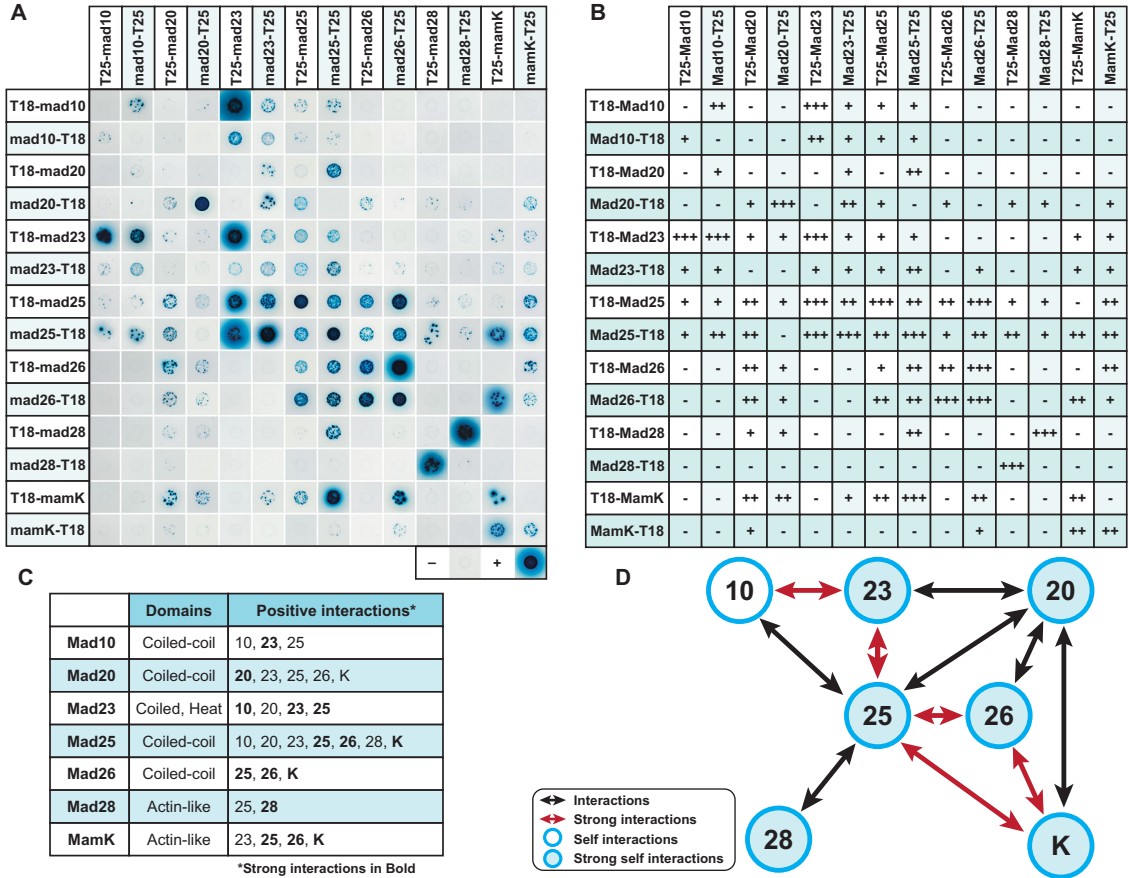

**Fig. 6 | Interactions of magnetosome chain organization proteins. A** Bacterial Adenylate Cyclase Two-Hybrid (BACTH) assay was conducted to assess interactions among magnetosome chain organization proteins—Mad10, Mad20, Mad23, Mad25, Mad26, Mad28, and MamK. Co-transformed *E. coli* DHM1 expressing T18- and T25-fusion proteins were spotted onto M63 maltose-mineral salts agar, where blue coloration indicates high β-galactosidase activity, signifying a positive interaction, while white indicates no interaction. The positive control (leucine zipper) resulted in dark blue spots, while the negative controls (empty vector combinations) produced white spots, as expected. **A** Photographic images of spots for each interaction combination. **B** A table summarizing all interaction combinations, scored based on results from **A** and categorized by color intensity: dark blue spots indicate strong interaction (+++), medium blue spots represent moderate (++), light blue spots signify minor (+), and white denotes no interaction (-). **C** A table highlighting the interactions. Strong interactions (+++) are indicated in bold. **D** A visual diagram illustrating these interactions: black arrows represent interactions, including bidirectional interactions (fusion with T25 and T18 interacting with their corresponding fusion partners), red arrows highlight stronger bidirectional interactions, open blue circles indicate observed self-interactions, and filled blue circles denote stronger self-interactions.

previously suggested to have an overlapping function with MamK[9,25] (Supplementary Fig. 1A). Surprisingly, deleting *mad28* in RS-1 resulted in a phenotype distinct from that of the Δ*mamK*. In Δ*mad28*, WT-like subchain organization was retained. However, chains were no longer exclusively localized to the positive cell curvature and were found pole-to-pole across the long axis at mid-cell, as well as organized diagonally or transversely across the cell (Fig. 5F, J, K and Supplementary Fig. 5G).

To further distinguish between the Δ*mad28* and Δ*mamK* strains, we examined chain formation in a biomineralization time-course experiment. In Δ*mad28*, chain assembly followed the WT pattern, with a single mature crystal forming first, followed by subchain development and crystal incorporation (Fig. 5I-K). However, chains were mispositioned from the start, forming pole-to-pole or diagonal/transverse orientations instead of localizing to the positive cell curvature as in WT (Fig. 5J, K). In contrast, Δ*mamK* exhibited a distinct chain formation pattern. Early biomineralization resembled WT, but as crystals matured, subchains failed to develop, resulting in a single continuous chain (Fig. 5L). Δ*mamK* mutants also had more magnetosomes per cell and a higher $C_{mag}$ than WT at both biomineralization stages (Fig. 5N, M, Supplementary Fig. 4B and D), a phenotype not observed in *mamK* mutants of AMB-1 and MSR-1[29,32]. These findings suggest that Mad28

aligns magnetosomes to the positive curvature early in chain formation, while MamK distributes subchains along the cell length, ensuring proper alignment and potentially regulating magnetosome production.

## Interaction Network of chain organization proteins
Bioinformatic analysis shows that Mad10, 20, 23, 25, and 26 contain coiled-coil domains which can facilitate a variety of protein-protein interactions[19] (Supplementary Fig. 1A, S8C-E). Mad23 also has HEAT repeat domains, known for similar functions (Supplementary Fig. 1A and 8 B). Additionally, MamK and Mad28 can form homo- and heteropolymers[25]. Thus, we hypothesized that these chain formation proteins may interact with one another. To explore potential interactions, we conducted Bacterial Adenylate Cyclase Two-Hybrid (BACTH) assays[33] (Fig. 6). We created N- and C-terminal fusions to T25 and T18 domains for each protein and tested all interaction combinations (Fig. 6 and Supplementary Fig. 9). Interactions were classified based on color intensity: dark blue spots as strong (+++), medium blue as moderate (++), and light blue as minor (+) (Fig. 6). Our results reveal an extensive and complex network of potential interactions. All proteins exhibited self-interactions in at least one fusion combination. Mad25 was the only protein that interacted with all other proteins tested,

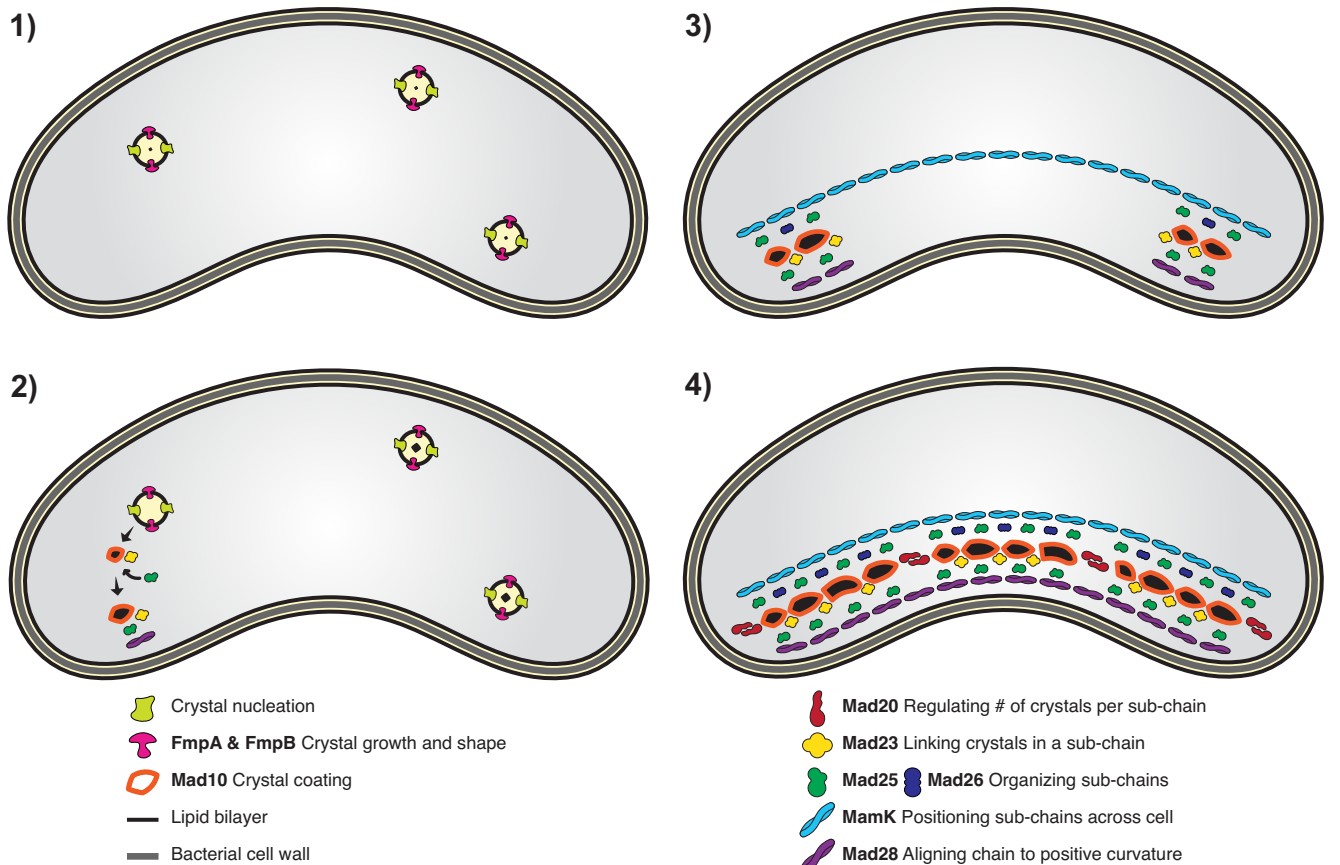

**Fig. 7 | Model of magnetosome synthesis and chain organization in RS-1.** Model of the different stages of biomineralization in *Desulfovibrio magneticus* RS-1. *Step 1 Nucleation and growth.* Biomineralization begins at nucleation sites randomly distributed within the cell. *Step 2 Crystal maturation and localization.* Maturing crystals are transported to the cell's positive curvature by a cohort of Mad proteins.

*Step 3 Subchain assembly.* As crystals reach the positive curvature, the process shifts to organizing them into subchains. *Step 4 Chain elongation and positioning.* Subchains are extended and aligned along the cell length by MamK, while Mad28 ensures the chain remains localized at the positive curvature.

while showing no interaction with the T18 and T25 negative controls (Fig. 6, Supplementary Fig. 9). Mad20, Mad23, Mad26, and MamK each interacted with three or more proteins. Mad10 interacted with itself, Mad23, and Mad25 (Fig. 6). Mad28 showed an interaction only with Mad25. Among these, Mad25 and Mad26, both containing extensive coiled-coil domains, displayed a particularly strong interaction. Likewise, Mad25 and Mad23 exhibited a strong interaction, likely driven by the coiled-coil domain of Mad25 and the HEAT domain of Mad23 (Supplementary Fig. 1A, 8B, and 8D). Despite the artificial nature of the BACTH assay, these findings provide further evidence for the existence of a distinct chain organization pathway in deep-branching MTB, dependent on the interactions of coiled-coil and actin-like proteins.

### A model for magnetosome formation in RS-1
The phenotypic and molecular data presented here suggest a new model for the temporal development of magnetosome chains in RS-1 with the following steps (Fig. 7):

Step 1: Nucleation and growth. Biomineralization initiates at randomly distributed sites within the cell. Mutations in *fmpA* and *fmpB* impair crystal maturation, reduce crystal numbers, and disrupt localization to the positive curvature, linking biomineralization to chain formation.

Step 2: Crystal maturation and localization. As crystals mature, they are transported to the cell's positive curvature. Mad10, previously shown to interact with magnetite[26], may facilitate this transition. Bacterial two-hybrid results suggest an assembly pathway where Mad10 interacts with Mad23 and Mad25 which mediate subsequent steps.

Step 3: Subchain assembly. Initially, one or two mature crystals appear at the positive curvature. Mad23, in conjunction of Mad25 and Mad26, links accumulating crystals to form and stabilize subchains.

Step 4: Chain elongation and positioning. In parallel to the previous steps, biomineralization sites are distributed along the cell length by MamK, while Mad28 ensures the localization of subchains to the positive curvature. Interestingly, RS-1 MamK and the deep-branching Mad28 can partially complement *mamK* loss in MSR-1[25]. However, our findings suggest that Mad28 mainly controls chain localization, while MamK orchestrates subchain positioning and the number of biomineralization events.

## Discussion
Magnetotactic bacteria and their magnetosome organelles present an ideal test case to uncover the mechanisms that underly the diversification of specific microbial traits. In recent years, an explosion of genomic and metagenomic studies have revealed that a core magnetosome gene set is often associated with accessory genes that are unique to specific phylogenetic families of MTB. Bioinformatic and structural prediction programs, as well as heterologous expression of these genes in established model systems, have not been successful in determining their role in magnetosome formation. Here, we deployed a new workflow to monitor the process of biomineralization in *D. magneticus* RS-1, a representative of deep-branching MTB, to reveal a distinct mode of organelle assembly mediated by a suite of accessory Mad proteins.

Several features of the RS-1 biomineralization and magnetosome chain formation process distinguish it from the model Alphaproteobacterial MTB. First, in RS-1 crystals are made serially, one or a few at a time. AMB-1 and MSR-1, however, produce multiple crystals in parallel[20,34]. Second, biomineralization is a pre-requisite for localization of crystals to the positive cell curvature in RS-1. In contrast, in AMB-1 magnetosome membranes are localized to the positive cell curvature even prior to the initiation of biomineralization[20]. And, while magnetic interactions play a role in chain assembly in MSR-1, mutants with severe biomineralization defects are still able to localize magnetosomes to the positive cell curvature[34–36]. Third, we notice a shift in the RS-1 magnetosome proteome across the stages of biomineralization. Since proteomic studies in AMB-1 and MSR-1 have been conducted at fixed timepoints, the quantitative shifts in their proteomes across biomineralization timepoints remain unknown. The closest parallels to our work with RS-1 come from a study in AMB-1 that separated magnetosomes based on their magnetic properties leading to the identification of MamY as a protein associated with smaller magnetite particles[37]. Additionally, proteomic comparisons of WT AMB-1 and a mutant lacking MamE protease activity show that despite a severe biomineralization defect the magnetosome protein content was essentially the same between the two strains[38]. An alternative to proteomics has been to visualize the localization of fluorescently tagged proteins in AMB-1 and MSR-1. While most proteins examined maintain a constant magnetosome localization, a handful of proteins, such as Mms6, MamD, and McaB, only localize to magnetosomes of AMB-1 after the initiation of biomineralization[39–41]. However, even these proteins seem to be at magnetosomes from the earliest stages of magnetite formation in clear contrast to the shifting proteome of RS-1 magnetosomes during the biomineralization process. Based on these circumstantial observations, we hypothesize that a similar proteomic transition does not occur in the Alphaproteobacterial MTB.

Notably, many of the proteins associated with the early biomineralization stages in RS-1 contain predicted transmembrane domains indicating the presence of a magnetosome membrane (Supplementary Fig. 1A). Indeed, previous chemical and UV mutagenesis of RS-1 identified transmembrane-domain-containing proteins such as MamB, MamL, MamQ, Mad2, Mad6, and FmpA as critical players in the early stages of biomineralization[12]. Magnetosome proteins enriched in later stages of RS-1 biomineralization do not contain predicted transmembrane domains and are instead enriched for coiled-coil and actin-like domains (Supplementary Fig. 1A). A global examination of the domain architecture of magnetosome proteins provides further evidence for the potentially distinct role of the magnetosome membrane in each MTB group. For instance, 65% of AMB-1 magnetosome proteins, but only 39% RS-1's, are predicted to contain transmembrane domains (Supplementary Fig. 10 A-B). However, approximately ~26% (10 proteins) of RS-1 magnetosome proteins contain predicted coiled-coil domains, in contrast to ~12% (6 proteins) of AMB-1's magnetosome proteins (Supplementary Fig. 10C and D). In fact, all of AMB-1's coiled-coil-domain-containing magnetosome proteins, and none of RS-1's, are predicted to contain transmembrane domains (Supplementary Fig. 10). This is consistent with previous findings that noted an absence of a magnetosome membrane around mature crystals in RS-1[15]. However, it is also possible that these late proteins are interacting with cytoplasmic domains of transmembrane domain proteins embedded in a magnetosome membrane that is hard to detect. In contrast, lipid-bilayer membranes surrounding mature magnetic particles can be readily observed in a variety of Alphaproteobacterial MTB[20,34,42].

Our genetic analysis of several *mad* genes, along with *mamK*, revealed a variety of chain formation defects. Based on the observed phenotypes and bacterial two-hybrid data, we hypothesize that these Mad proteins promote a series of homo- and/or hetero-polymeric reactions to assemble individual magnetic particles into subchains and separate subchains across the long axis of the cell. Notably, other than MamK, all known magnetosome chain formation factors of AMB-1 and MSR-1 (MamJ, MamY, LimJ, MamK-like, McaA, McaB, and MamF and its homologs) are absent in RS-1[28,34,41,43–45]. Similarly, MTB of the deep-branching *Elusimicrobiota* phylum members which do not form organized magnetosome chains are missing MamK as well as the chain formation Mad proteins. While genome incompleteness should be considered, the two *Elusimicrobiota* genomes show completeness levels of 75.84% and 94.38% in GTDB[8,46]. Therefore, we suggest that these coiled-coil proteins along with the actin-like MamK and Mad28, represent a mechanistic mode of chain organization exclusive to deep-branching MTB.

Although the specific proteins involved in chain organization in RS-1 differ from those of Alphaproteobacterial MTB, similar deletion phenotypes can be observed across these distantly related species. For example, deletion of *mad10* in RS-1 produces a phenotype mirroring that of *mamJ* deletion in MSR-1 and the triple deletion missing *mamJ*, *limJ*, and *mcaA* in AMB-1[41]—all resulting in magnetosome clustering. Similar to Mad10, MamJ was originally hypothesized to be involved in biomineralization due to the presence of a highly acidic domain[34]. However, this domain is not involved in chain organization activity of MamJ and its ability to bind magnetite is unknown[47]. Other mutants also carry superficial phenotypic similarity between RS-1 and AMB-1/MSR-1. For instance, deletion of *mamK* in RS-1 and the loss of *mcaA/mcaB* in AMB-1 lead to loss of subchains in both organisms[41]. The scattered magnetosomes of Δ*mad23* mutant of RS-1 also resemble the chain formation defects of the Δ*F3* mutant of MSR-1[43]. These findings indicate that non-homologous proteins may fulfill analogous roles in different MTB groups. They also suggest that disruptions in the organization of a chain of magnetic particles can lead to similar final outcomes.

Collectively, our findings shed further light on the evolution and diversification of magnetosomes. Nearly all MTB share genes involved in membrane formation (*mamQ*), iron transport (*mamB*), and chain organization (*mamK*), likely forming the foundations of an ancient iron-accumulating organelle. Diverse magnetosome phenotypes, such as crystal shape variations, may have emerged through addition of subgroup-specific proteins to this core system. In our model, Alphaproteobacterial MTB developed a system to align magnetosome membranes into chains using Mam proteins, while RS-1 and other deep-branching MTB evolved to organize biominerals into chains using Mad proteins. This evolutionarily distinct solution may have arisen to combat the previously described conflict between magnetocrystalline orientation of tooth-shaped magnetosomes and their function as a navigational tool[48,49].

## Methods

### Multiple-sequence alignments and tree construction

Dataset was designed by searching IMG/MER[50] for all known MTB species with homologs for all five of the core *mam* genes selected: *mamA, mamB, mamE, mamI* and *mamQ*. Next, the dataset was refined to encompass only species with available genomes and accessible TEM images, enabling the identification of magnetosome gene sets (*mam*, *mms*, *mad*, *man* and *mae*) for each species, along with their respective crystal shapes and compositions. Supplemental Table 1 is a list of all species included in designing the phylogenetic tree for this analysis. Once the dataset was complete, the 16S DNA sequences for each species were obtained from NCBI or IMG-MER[50] (Supplemental Table 1). Sequences were aligned using MAFFT version 7.490 using the scoring matrix 1PAM/ k = 2[51]. Next, the aligned sequences were used to generate the maximum likelihood tree using PhyMLTree version 3.3.20180621 using GTR substitution model with 1000 bootstraps[52]. Additionally, the tree was rooted with *Omnitrophus magneticus* SKK-01. FigTree v1.4.4 was used to visualize tree and was then it was exported to Illustrator to generate Fig. 1.

## General culturing for RS-1

*Desulfovibrio magneticus* RS-1 strains were grown at 30 °C anaerobically in RS-1 growth medium (RGM), as described previously[12,15]. For growth with hydrogen, the medium was gassed with 10% hydrogen balanced with Nitrogen prior to autoclaving the medium. Additionally, after inoculating, the headspace was re-gassed with the same concentration of hydrogen/nitrogen gas mixture for 10 min. All hydrogen cultures were grown spinning on a wheel in the 30 °C incubator. Cultures with nitrogen in the headspace that were used as controls for these hydrogen cultures were grown on a wheel in the incubator. All other nitrogen cultures were grown without shaking. All cultures were prepared in triplicate, using separate colonies to generate three biological replicates.

For experiments with different concentrations of hydrogen (Fig. 2A), RS-1 WT strain was grown at 30 °C anaerobically in RS-1 growth medium (RGM), as described above. Initially, all culture tubes were purged with nitrogen gas. Subsequently, prior to inoculation, varying volumes of nitrogen from the headspace were displaced and substituted with an equivalent volume of 100% hydrogen gas, aiming to achieve distinct concentrations of hydrogen in the headspace. Afterwards, RS-1 was inoculated into the culture tubes. Optical density and magnetic response via $C_{mag}$ was monitored throughout growth as previously described[12] using a Spectronic 20D+ Spectrophotometer.

## Culturing for proteomics

Cultures for hydrogen vs. nitrogen consisted of 350 mL cultures with 2/3 volume of headspace, which were prepared by gassing either with nitrogen or 10% hydrogen and grown by shaking at room temperature. At late exponential phase, cultures were harvested by centrifugation at 8,000 x g using the Beckman Model J2-21M/E centrifuge. Cell pellets were stored at −80 °C until lysis and preparation for proteomic analysis. To examine different stages of biomineralization, 300 mL initial cultures were grown with 10% hydrogen in the headspace to remove all magnetosomes. $C_{mag}$ was measured as described previously[12] and TEM images were taken of initial cultures to ensure no magnetosomes were present. Then 10 1 L bottles of RGM-X medium were prepared by infusing the medium with nitrogen and gassing the headspace with nitrogen. Each of the 3 biological replicates consisted of 10 1 L bottles, which were inoculated with the same starting culture. Three bottles were used for late biomineralization, which were inoculated with 10 mL of starting culture, and seven bottles were used for early biomineralization samples, which were inoculated with 20 mL of the starting culture. The cultures were grown in a 30 °C incubator and OD and $C_{mag}$ were taken 2X daily. When cultures reached a $C_{mag}$ of 1.05–1.15 seven of the bottles were harvested for early biomineralization samples by centrifugation at 8,000 x g. Subsequently, when the final 3 bottles had a $C_{mag}$ between 1.30 and 1.5 they were harvested for late stages of biomineralization samples also by centrifugation at 8,000 x g. Cell pellets were stored at −80 °C until it was time to prepare samples for proteomics. Each biological replicate originated from a single colony, which was inoculated into 300 mL pre-culture bottles grown with hydrogen. Each 300 mL pre-culture was subsequently used to inoculate a set of ten 1 L bottles, resulting in three 10 L cultures derived from three separate colonies.

## Collecting proteins associated with magnetosomes using a magnetic column

After harvesting cultures, cell pellets were resuspended in buffer A (25 mM Tris, pH 7.0, 100 mM sucrose, 1 μg/mL Leupeptin, 1 μg/mL Pepstatin A, and 1 mM PMSF. Cells were lysed three times at 25 kpi using the Single Shot feature of the Multi Cycle (MC) Cell Disruptor from Constant Systems[53]. Magnetic LS Columns (130-042-401) from Miltenyi Biotec were used to separate the magnetosomes from the cell lysate. Magnetic columns were primed with a wash with buffer A, followed by the attachment of magnets to the outside of the columns. Subsequently, the cell lysate was passed through the column by gravity flow. Flow through fractions were passed over the column two more times and buffer A was used for three additional washes. Following wash cycles, the magnets were detached from the column, and 500 μL of buffer A was pipetted into the column. The plunger provided with the column was then to provide additional force to collect all of the magnetosomes. Some of the cell lysate from each sample was also saved as a control. Next, a Bradford assay was conducted on all samples to determine the protein concentration, and if necessary, samples were concentrated during this stage. Finally, all samples were trypsin-digested as previously described[21] to prepare them for liquid chromatography-mass spectrometry, as detailed below.

## Liquid chromatography-mass spectrometry

Trypsin-digested protein samples were each analyzed in triplicate using an Acquity M-class ultra-performance liquid chromatography (UPLC) system that was connected in line with a Synapt G2-Si mass spectrometer (Waters, Milford, MA) as described elsewhere[21]. Data acquisition and analysis were performed using MassLynx (version 4.1, Waters) and Progenesis QI for Proteomics software (version 4.2, Waters Nonlinear Dynamics). Data were searched against the *Desulfovibrio magneticus* strain RS-1 translated protein database to identify peptides[54].

## Plasmid and cloning and deletion

All plasmids used in this study for generating deletions and for their complementations in RS-1, as well as for plasmids used in the BACTH assays, are listed in Supplemental Table 3. In-frame deletion vectors targeting *mad10, mad20, mad23, mad25, mad26, mad28* and *mamK* were constructed by amplifying upstream and downstream homology regions of RS-1 genomic DNA, using the primers listed in Supplementary Table 4. The homology regions were then inserted into the XbaI site of pAK1127 using the Gibson cloning method. The P*npt* -*strAB* cassette was subsequently inserted between the upstream and downstream homology regions of the deletion vector. Plasmids were transformed into *Escherichia coli* WM3064 and then transferred to *D. magneticus* RS-1 using conjugation methods as described previously[13]. Allelic replacement of *mad10, mad20, mad23, mad25, mad26, mad28* or *mamK* with *strAB* was achieved with streptomycin selection and using two counterselections, 5-Flurouracil and sucrose. All deletions were confirmed by PCR followed by full genomic Illumina sequencing using SeqCenter (https://www.seqcenter.com).

To generate plasmids used to complement deletion strains, the genes *mad10, mad20, mad23, mad25, mad26, mad28* and *mamK* were individually amplified using primers listed in Supplemental Table 4 and were cloned into pAK906 (containing the *npt* promoter) and pAK907 (containing a *mamA* promoter) using Gibson assembly. Both complementation plasmids for each gene were tested by transformation into the respective RS-1 deletion strain. Each strain, containing different versions of the complementation plasmid, was grown in liquid culture and evaluated by $C_{mag}$ measurements and TEM to determine which provided the best complementation. The plasmid that most effectively restored the phenotype was selected for further experiments.

To generate plasmids used in the BACTH studies, *mad10, mad20, mad23, mad25, mad26, mad28* and *mamK* were either synthesized by Twist Bioscience (https://www.twistbioscience.com) or PCR amplified using the primers listed in Supplementary Table 4 and were cloned into pKT25 or pKNT25 (the N or the C-termini of the T25 fragment) and pUT18 or pUT18C (the N or the C-termini of the T18 fragment) vectors in frame with the T25 and T18 fragment open reading frames by Gibson assembly.

## Biomineralization time course using hydrogen

RS-1 strains were grown at 30 °C anaerobically, as described above. Initial cultures were all grown with medium infused with 10% hydrogen and headspace was purged again with 10% hydrogen after inoculation. All hydrogen cultures were grown on a wheel in the incubator. Additionally, all cultures were transferred for supplementary growth into hydrogen tubes to ensure the removal of all magnetosomes. Varying volumes (1 mL, 0.5 mL, 0.2 mL, and 0.1 mL) of hydrogen grown cultures were transferred into nitrogen-infused medium, aiming to capture all stages of biomineralization more effectively. Optical density and magnetic response via $C_{mag}$ was monitored throughout growth as previously described[12] and at different stages of biomineralization samples were saved for TEM imaging. The progression of biomineralization stages was tracked using $C_{mag}$, with values ranging from 1.05 to 1.10 classified as early stages, 1.15 to 1.20 as middle stages, and 1.3 to 1.6 as late stages. All biomineralization experiments were done with three biological replicates and individual TEM grids were generated for each biological replicate.

## Bacterial adenylate cyclase two-hybrid (BACTH) assays

The assay was performed as described in the Euromedex BACTH system kit manual. N- and C-terminal T18 and T25 fusions of Mad10, Mad20, Mad23, Mad25, Mad26, Mad28 and MamK proteins were constructed using plasmid pKT25, pKNT25, pUT18C, and pUT18 in *E. coli* K12 strain DH5-alpha. All plasmids were sequence-verified to show T18/T25 magnetosome protein fusions were correct. The fusions were then co-transformed into competent *E. coli* DHM1 cells (lacking endogenous adenylate cyclase activity) in all pairwise combinations, plated on LB agar plates containing 100 µg/mL carbenicillin and 50 µg/mL kanamycin, and incubated at 30 °C overnight. Several colonies of T18/T25 co-transformants were isolated and grown in LB liquid medium with 100 µg/mL carbenicillin and 50 µg/mL kanamycin overnight at 30 °C with 220 rpm shaking. Overnight cultures were spotted on indicator plates containing minimal M63/maltose medium with agar supplemented with 40 µg/mL X-gal, 50 µg/ml carbenicillin; and 25 µg/ml kanamycin. Plates were incubated 4 to 8 days at 30 °C before imaging. Bacteria expressing interacting hybrid proteins will show blue, while bacteria expressing non-interacting proteins will remain white. Different concentrations of IPTG (0, 0.1 mM and 0.5 mM) were tested in the overnight culture as well as on the indicator plates with little to no difference. Therefore, IPTG was not included in the final cultures and plates of this experiment. Plates were imaged on a light box using a Canon EOS R50 camera with a Canon RF35mm F1.8 Macro STM Lens.

## Electron microscopy

For TEM analysis each strain of RS-1 was cultivated in 10 mL of RGM medium anaerobically and cells were collected for imaging at mid-exponential phase of growth. The 1 mL of cells collected were pelleted and resuspended into 10 µL of RGM medium. The resuspended cells were applied on a 300-mesh copper grid coated with Formvar and carbon films (Electron Microscopy Sciences) that were glow discharged for 10 seconds using the PELCO easiGlow. After two minutes on the grid the remaining droplet was wicked away with filter paper. Grids were then washed three times with milliQ water and remaining water was wicked away with filter paper after each wash. Then the cells were imaged on an FEI Tecnai 12 transmission electron microscope equipped with a 2k × 2k charge-coupled device camera (The Model 994 UltraScan®1000XP) at an accelerating voltage of 120 kV using Gatan Digital Micrograph at the EM-Lab at the University of California, Berkeley (https://em-lab.berkeley.edu/EML/index.php). Magnetosomes were quantified and measured manually using ImageJ 1.54 g. Microsoft Excel v 16.95.4 was used to record all data. The statistical analysis of magnetosomes per cell and chain or subchain organization in each mutant was performed using GraphPad Prism (version 9 and version 10) statistical software.

## Reproducibility

All TEM micrographs are representatives of the strains grown under the specified conditions. Consistent magnetosome chain organization was observed in both WT and RS-1 mutant cells across at least three independent experiments. Likewise, similar magnetosome shape and distribution in the *fmpA* and *fmpB* mutants (e.g., Fig. 4) were observed in over three independent experiments.

## Reporting summary

Further information on research design is available in the Nature Portfolio Reporting Summary linked to this article.

## Data availability

All data are available within in the article and supplementary files. The proteomic datasets generated and analyzed during the current study are available in the MassIVE repository, (identifier: MSV000099525). All data used to generate figures and supplemental figures are available in the associated Source Data file. Source data are provided with this paper.

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

## Acknowledgements

A.K. and V.V.R. are supported through the National Institute of General Medical Sciences (R35GM127114). The QB3/Chemistry Mass Spectrometry Facility at the University of California, Berkeley received support from the National Institutes of Health (grant 1S10OD020062-01). Additionally, V.V.R. was supported through the Genetics Training Grant, which receives funds from the National Institutes of Health (Grant #: 1T32GM132022). Thank you to the staff at the University of California Berkeley Electron Microscope Laboratory for advice and assistance in electron microscopy.

## Author contributions

V.V.R. conceived, performed, and analyzed all the experiments. A.T.I. ran and preformed initial analysis of the proteomics data for both proteomics experiments. E.O. generated the ΔmamK deletion mutant and C.R.G. generated the Δmad23 deletion mutant. A.K. supervised the experimental design, data analysis, and data presentation. V.V.R wrote the manuscript. A.K. assisted in editing and revision of the manuscript. All co-authors edited the manuscript.

## Competing interests

The authors declare no competing interests.
