## [Peer Review File · Nature Communications]

A network of coiled-coil and actin-like proteins controls the cellular organization of magnetosome organelles in deep-branching magnetotactic bacteria

Corresponding Author: Professor Arash Komeili

Version 0:

Reviewer comments:

Reviewer #1

(Remarks to the Author)

In their manuscript Russell et al. combine genetic and induction experiments with proteomic and bacterial two-hybrid analyses to investigate magnetosome chain formation in the deep-branching magnetotactic bacterium *Desulfovibrio magneticus* RS-1. The authors report that magnetic particles initially form randomly within the cell before localizing to the cell's positive curvature. While early biomineralization involves membrane-associated proteins conserved across all MTB, later stages depend on coiled-coil proteins (Mad20, 23, 25, and 26) and actin-like proteins (MamK and Mad28). The study presents a series of well-executed experiments that mostly support the overall conclusions. As such, the study is one of the first to analyze the function of several conserved Mad proteins. Therefore, the results are of special interest for the understanding of magnetosome chain formation within deep-branching MTB and a valuable contribution to literature. However, as will be outlined below the manuscript should be improved in several areas before publication.

Major comments:

The discussion places too much emphasis on the newly proposed model. While the model is interesting and consistent with the observed results, it remains highly speculative at this stage, as alternative (including temporal) models could also be valid. Therefore, in the absence of additional supporting evidence, a more cautious and balanced interpretation is strongly recommended.

Equally important, the discussion provides only limited comparison with existing literature, which makes it challenging to fully assess the novelty and significance of the study's findings. For instance, it would be helpful to include at least a few sentences addressing how the newly proposed model relates to previous models of magnetosome formation in RS-1 (eg as discussed in reference 12). Additionally, many RS-1 mutants appear to phenocopy deletion strains of alphaproteobacterial MTB (eg $\Delta mamK_{RS-1} \sim \Delta mcaAB$; $\Delta mad10 \sim \Delta mamJ$; $\Delta mad28 \sim \Delta mamY$; $\Delta mad23 \sim \Delta F3$). Although the specific proteins and mechanisms underlying magnetosome chain formation in RS-1 differ from those in alphaproteobacterial MTB, they seem to address similar biological challenges and fulfill comparable functions. This parallel should be more explicitly discussed and critically compared in the manuscript.

Several of the proteins analyzed in this study contain coiled-coil domains. While the authors speculate that these proteins may form intermediate filaments (ll. 324–325), coiled-coil domains are known to serve a variety of functions beyond filament formation. Therefore, it would be important to clarify why the authors favor the hypothesis of filament formation over alternative functional roles. Furthermore, to provide better context for this interpretation, a brief paragraph summarizing the biological roles and versatility of coiled-coil domains should be included in the introduction.

This may be a matter of personal preference, but differential proteomics data is commonly presented as \log_2 fold changes (eg 10.1038/s41586-021-04003-2). More importantly, however, it is crucial to provide access to the complete proteomic dataset to ensure transparency and avoid the impression that only data supporting the authors' hypotheses has been selectively presented. Currently, it remains unclear whether other MGC-encoded proteins were also detected in the magnetosome-enriched or cell lysate fractions. Were these proteins identified but showed no significant difference between conditions, or were they not detected at all? I strongly recommend including a volcano plot, as shown in Ref. 10.1038/s41586-021-04003-2, to visualize not only the fold changes but also the statistical significance of the results, which is currently missing from the manuscript.

The arrangement and organization of the figures could be improved, as they often do not follow the logical flow of the text (eg Fig. 2B is referenced after Figs. 2C and 2D). Although this is not a critical issue, it makes the manuscript more difficult to

follow, particularly when multiple images are referenced in quick succession (eg ll. 195–201).

Minor comments:

ll. 23-25: “These findings suggest that while biomineralization originates from a common ancestor, magnetosome chain formation has diverged evolutionarily among different MTB lineages.” – This is not a novel finding of the current study. Simple genome comparisons lead to this conclusion previously (e.g. 10.1093/nsr/nwac238). The sentence should therefore be rephrased.

l. 43 “*Hydrpgendentota*” – correct to “*Hydrogenedentota*”

ll. 54 “Among these, only five core genes (mamA, B, I, E, and Q) are shared across all MTB^{8,9}.” – Reference 9 shows more than 5 conserved genes. Moreover, while the absence of magnetosome chains in Elusimicrobiota seem to support the absence of MamK, the genome sequences should be treated with caution due to incompleteness (ref 8).

ll. 56 “mad (magnetosome deep branching)” genes - have been originally named ‘magnetosome associated Deltaproteobacteria’ in 10.1111/1462-2920.12128

l. 57 „man (magnetosome nitrospirota)” genes - have been originally named ‘magnetosome-associated *Nitrospirota*’ in 10.1186/s40168-024-01837-6

ll. 67 “(Supplemental Figure S4E, Figure 2D-ii).” – Figure S4E does not exist!

l. 144 “revealed two mature magnetosome morphologies: curved and straight (Figure 3F ix and x).” – The magnetosomes at early biomineralization time points cannot be distinguished between curved and straight crystals. The panel is thus misleading to show early steps of curved magnetite crystals. Additionally, how can the authors be sure that not all crystals are curved when only looking at them in 2D-TEM?

ll. 150 “In contrast, Alphaproteobacterial MTB such as AMB-1 and MSR-1, localize magnetosome membranes to the positive curvature before biomineralization begins^{2,21}. – Ref 2 is not appropriate for MSR-1. Additionally, in 10.1038/nature04382 it has been described that magnetosome formation in MSR-1 occurs throughout the cell! “Alphaproteobacterial” should not be italicized.

ll. 182 “In addition to the defects in crystal shape and size, *fmpA* and *fmpB* mutants exhibited abnormal crystal placement with most crystals not localized to the cell’s positive curvature (Figure 4J).” - Ref. 12 and exemplary pictures in Fig 4A and B show mainly localization to the positive curvature and thus question the conclusion.

ll. 184 “The crystals of the *fmpA* mutant also fail to form chains, while only 5% of *fmpB* mutant cells show chain organization (Figure 4I).” - What is considered a chain? Isn’t it obvious to have almost no chain formation with just 1 to 3 particles? In Ref. 12 chain formation is shown, indicating that chain formation is generally still possible.

ll. 187 “We propose that crystal nucleation occurs throughout the cell and crystals are later transported to the positive curvature post-maturation.” – What means post-maturation here? In TEM images one can also see immature crystals at the positive curvature. Please describe more specifically.

l. 193 “Mad10 is more abundant in early-stage magnetosome samples than in late stages ...” – Is this change significant. In Fig. S8A one can get the impression that there seem to be similar levels of Mad10 at magnetosomes.

l. 195 “Previous studies identified Mad10 as a magnetite-binding protein and hypothesized a role in magnetite nucleation or shape control.” – Similar assumptions have been made for MamJ (10.1038/nature04382). This might be a nice point to discuss.

ll. 214 “In the $\Delta mad23$ mutant, single magnetosomes are dispersed, with the fewest crystals per subchain among mutants and WT.” – Here it should also be mentioned if the overall number of particles in the mutant is similar to WT or also reduced.

ll. 216 “The strain also has the lowest CMag of all chain organization mutants (Supplemental Figure 1B), suggesting Mad23 connects individual magnetosomes to form subchains.” – According to Fig. S1C $\Delta mad10$ has even lower CMag! How do you explain these discrepancies? How do the authors distinguish between the connection of magnetosomes into subchains versus the formation of a continuous chain in this case? In other words, the $\Delta mamK$ mutant also lacks subchains but still assembles a magnetosome chain, suggesting that subchain formation is not essential for chain formation. This implies that the role of Mad10 likely extends beyond merely promoting subchain formation.

ll. 228 “In AMB-1, for example, TEM shows subchains of magnetic particles, separated by empty magnetosome membranes all of which are organized into a continuous chain.” – Empty vesicles are usually not visible by conventional TEM.

ll. 249 “However, chains were no longer exclusively localized to the positive cell curvature and were found pole-to-pole at mid-cell, ...” – For better clarity, the manuscript should explicitly specify which cellular axis is meant when referring to the “mid-cell” position.

ll. 277 “Mad25 interacted with all tested proteins, while Mad20, 23, 26, and MamK interacted with three or more.” – Please describe in more detail.

ll. 287 “Our work suggests a new model of magnetosomes formation with the following steps (Figure 6):” – This sentence implies that the authors describe a new model for magnetosome formation that is valid for all MTB. However, the findings here are restricted to deep-branching MTB. The sentence should therefore be modified to clarify this fact.

ll. 289 “Biomineralization initiates at randomly distributed membrane bounded nucleation sites that are likely membrane-enclosed.” – On which data or references are these assumptions based, particularly regarding the membrane-bound nucleation?

ll. 290 “Mutations in *fmpA* and *fmpB* impair crystal maturation, reduce crystal numbers, and disrupt localization to the positive curvature, linking biomineralization to chain formation.” – As mentioned above: Ref. 12 and exemplary pictures in Fig 4A and B show mainly localization to the positive curvature and sometimes chain formation. The conclusions should thus be discussed more critically.

ll. 296 “Bacterial two-hybrid results suggest an assembly pathway where Mad10 interacts with Mad23, which subsequently connects to Mad25.” - There is also direct Mad10-Mad25 interaction (Fig. S2A and D)!

ll. 313 “AMB-1 and MSR-1 maintain a largely constant proteome^{2,33,34}. – Why mixing a review and primary literature here? References 33,34 are not proteomic studies and tested localization of few selected fluorescently labeled magnetosome proteins. The statement should therefore be expressed more carefully.

II. 320 "Additionally, other than MamK, all known magnetosome chain formation factors of AMB-1 and MSR-1 (MamJ, MamY, LimJ, MamK-like, McaA, and McaB) are absent in RS-1." - If even orthologs of MamJ and MamK are included among the factors mediating magnetosome chain formation, why are MamF-like proteins not mentioned?

II. 334 "This evolutionarily distinct solution may have arisen to combat the previously described conflict between magnetocrystalline orientation of tooth-shaped magnetosomes and their function as a navigational tool." – Could this also simply be due to the increased magnetic forces imposed by the larger magnetite crystals?

II. 342 "... homologs for all five of the core mam genes selected: *mamK*, *mamA*, *mamB*, *mamM* and *mamQ*." – The selected genes differ from those described in the introduction (*mamA*, *B*, *I*, *E*, and *Q*, I. 54). Explain why.

Figures 1 and S4: Different versions of the RS-1 MGC in different figures. Why?

Figure 1B: update nomenclature. Among others: LM-1 is known as *Candidatus Magnetocavibrio boulderlitore* LM-1 10.1038/s41396-020-0647-x

Figure S3A: Which MamEO is shown here (C-term or N-term)? In Figure S4 MamEO-N-term is shown to have transmembrane domains.

Figure S7D: The WT is shown to have dispersed magnetosomes almost exclusively!?

Cells in TEM images often appear pretty dark which makes the identification of magnetosomes difficult.

Reviewer #2

(Remarks to the Author)

Dr. Komeili's team has once again presented remarkable advances in the cell biology of magnetotactic bacteria. Russel et al. elegantly demonstrated the significance of Mad proteins in the formation and organization of magnetosomes in a magnetotactic bacterium from the Desulfobacterota phylum. They also proposed an interesting model for biomineralization in deep-branching magnetotactic bacteria (MTB). I was delighted to read the manuscript. However, I have a few minor comments:

The authors should check the order of the supplemental figures and tables, along with the citations in the main text. For instance, I could not find citations for Supplemental Tables 5-8 within the main text.

Regarding scale bars, there seems to be inconsistency in their color, with some in black and others in white. The black scale bars are more visible (for example, in Fig. 2, Fig. 3F, 3H, Fig. 4A, and Fig. 4B). Additionally, the font size appears to differ in Fig. 2G i, ii, and iii compared to other scale bars in Fig. 2. Furthermore, scale bars are missing in Fig. 3A and 3B.

In the sentence, "Other sets of conserved genes, called mad (magnetosome deep branching) are found in all deep-branching MTB; man (magnetosome nitrospirota) genes are found in Nitrospirota, and mae (magnetosome elusimicrobiota) genes are found in Elusimicrobiota8," please include the reference for the mad and man genes. The abbreviation "mad" was initially used to refer to "magnetosome-associated Deltaproteobacteria" rather than "magnetosome deep branching." Are you proposing an updated abbreviation or do you have a reference for this change?

Regarding updates, it would be interesting to mention after the sentence about RS-1 strain classification (lines 64-65) that this bacterium was previously classified as Deltaproteobacteria because of the references.

In the last sentence discussing the hydrogen effect on magnetosome synthesis (lines 117-118), it would be interesting to add the conclusion presented in the extended results section.

In the Electron Microscopy section, please consider adding a sentence indicating that statistics for magnetosome measurements are displayed in Tables 5-8.

When scale bars are the same across a set of images (for example, Fig. 2E, Fig. 3F, Fig. 3H, Fig. 4G, Fig. 4H, and others), the authors could include just one scale bar and specify in the figure legend that this scale bar applies to the entire set of images. This approach may help clarify the figure by reducing visual elements. However, if the authors prefer to keep individual scale bars, note that scale bars are missing in Fig. 3F iv and viii.

In Fig. 6, do the black circles in 1 and 2 represent a double-layered membrane? If so, the membrane should also have a label in the schematic representation.

On line 765, "E. coli" should be italicized.

Lastly, in Supplemental S7, there is a missing space between the words "ChainPlacement."

Arash Komeili
Professor
Plant and Microbial Biology
College of Natural Resources

261 Koshland Hall, MC 3102
Berkeley, CA 94720-3012
510 642-2217 phone
510 642-4995 fax
komeili@berkeley.edu

Thank you for the overall positive feedback on our manuscript. Based on your comments, we have extensively revised the text. Our point-by-point responses are below in red and we are happy to address any further concerns.

Reviewer #1 (Remarks to the Author):

In their manuscript Russell et al. combine genetic and induction experiments with proteomic and bacterial two-hybrid analyses to investigate magnetosome chain formation in the deep-branching magnetotactic bacterium *Desulfovibrio magneticus* RS-1. The authors report that magnetic particles initially form randomly within the cell before localizing to the cell's positive curvature. While early biomineralization involves membrane-associated proteins conserved across all MTB, later stages depend on coiled-coil proteins (Mad20, 23, 25, and 26) and actin-like proteins (MamK and Mad28). The study presents a series of well-executed experiments that mostly support the overall conclusions. As such, the study is one of the first to analyze the function of several conserved Mad proteins. Therefore, the results are of special interest for the understanding of magnetosome chain formation within deep-branching MTB and a valuable contribution to literature. However, as will be outlined below the manuscript should be improved in several areas before publication.

We thank the reviewer for their thorough review. In particular, their suggestions on how to present the significance of the results and their relationship to the existing literature have helped us formulate a clearer discussion of our work.

Major comments:

The discussion places too much emphasis on the newly proposed model. While the model is interesting and consistent with the observed results, it remains highly speculative at this stage, as alternative (including temporal) models could also be valid. Therefore, in the absence of additional supporting evidence, a more cautious and balanced interpretation is strongly recommended.

We have altered the discussion in the following ways in order to address this comment as well as others raised by both reviewers. Here are the major changes:

1. We moved the model to a dedicated section right before the discussion section (lines 305-322). It should be noted that we are describing a temporal model (at least temporal across biomineralization induction experiment). Additionally, we simplified some of the statements to be more directly reflective of results in this and previous papers.
2. The discussion is now focused on comparing RS-1 and AMB-1/MSR-1. The organization is as follows:
 - **Paragraph 1:** Address need for model system studies in deep-branching MTB
 - **Paragraph 2:** Compare/contrast biomineralization features of RS-1 and AMB-1/MSR-1. These include serial vs. parallel production of magnetosomes, link between biomineralization and subcellular localization of magnetosomes, and shifts in the magnetosome proteome during biomineralization.
 - **Paragraph 3:** Discussion of the role of magnetosome membrane in various MTB groups.
 - **Paragraph 4:** Chain formation factors in RS-1 vs. AMB-1/MSR-1.

- **Paragraph 5:** Similarities in chain formation defects.
- **Paragraph 6:** Conclusion, based on preceding paragraphs, that chain organization has a distinct evolutionary path in various MTB groups.

Equally important, the discussion provides only limited comparison with existing literature, which makes it challenging to fully assess the novelty and significance of the study's findings. For instance, it would be helpful to include at least a few sentences addressing how the newly proposed model relates to previous models of magnetosome formation in RS-1 (eg as discussed in reference 12). Additionally, many RS-1 mutants appear to phenocopy deletion strains of alphaproteobacterial MTB (eg $\Delta mamK_{RS-1} \sim \Delta mcaAB$; $\Delta mad10 \sim \Delta mamJ$; $\Delta mad28 \sim \Delta mamY$; $\Delta mad23 \sim \Delta F3$). Although the specific proteins and mechanisms underlying magnetosome chain formation in RS-1 differ from those in alphaproteobacterial MTB, they seem to address similar biological challenges and fulfill comparable functions. This parallel should be more explicitly discussed and critically compared in the manuscript.

We thank the reviewer for this excellent observation. We agree that there are general similarities between our mutants and known chain formation mutants of AMB-1 and MSR-1. What is fascinating is that unlike the shared Mam proteins, nearly all of the specific proteins involved in chain formation are not conserved amongst these organisms. There are also important phenotypic differences between the various mutants. Here are some of our thoughts on the mutants highlighted by the reviewer:

1. The $\Delta mad10$ mutant has phenotypic similarities to the $\Delta mamJ$ mutant of MSR-1 and the $\Delta mamJ\Delta limJ\Delta mcaA$ mutant of AMB-1. The fact that both MamJ and Mad10 were first proposed to be involved in biomineralization further enhances these parallels. We have added this to the discussion in lines 396-399.
2. The $\Delta mcaA\Delta mcaB$ strain of AMB-1 has some similarities to $\Delta mamK$ strain of RS-1. But, there are important differences as well because $\Delta mamK$ has significantly higher number of magnetosomes per cell and the chain is not centered in the cell (See Figure 5, Supplemental Figure S4D, and S5 Fii). The $\Delta mcaA\Delta mcaB$ strain of AMB-1 does not have defects in centering and segregation of the chain and essentially resembles WT MSR-1. This is because McaAB directs the placement of new magnetosomes membranes in between preexisting magnetosomes. Without them new magnetosomes are formed at the ends of the chain.
3. There are also key differences between $\Delta mad28$ and $\Delta mamY$. In $\Delta mamY$ the chain localizes to the negative curvature. In $\Delta mad28$ the chain is seen in many different orientations including the middle of the cell across its long axis or crossing the cell transversally (See Figure 5 and Supplemental Figure S7). These phenotypes suggest that the crystals are not necessarily localized to the cell membrane in the absence of Mad28.
4. The $\Delta mad23$ and $\Delta F3$ phenotypic similarities are also intriguing but more detailed studies are absent and thus make it difficult to link through a shared mechanistic pathway. There are also

important differences between these mutants since many of the crystals in the *Δmad23* mutant are part of a small subchain of 2-3 crystals whereas the magnetosomes of the *ΔF3* mutant appear to be isolated from one another.

We believe that these phenotypic similarities are fully consistent with the independent evolutionary paths of chain organization in different groups of MTB. Biophysical properties of magnetic particles means that they will likely aggregate in the absence of some chain assembly factors (Mad10/MamJ). Similarly, to function in orienting the cell efficiently, magnetosome chains have to be localized to the positive cell curvature in which case loss of MamY/Mad28 leads to the same general phenotype. We have added a section in the discussion to address these points (lines 392-405). Please note that our response in this letter is more thorough than what is written in the discussion section. This is primarily to balance the space given to all aspects of the discussion. We are happy to make changes as requested.

Several of the proteins analyzed in this study contain coiled-coil domains. While the authors speculate that these proteins may form intermediate filaments (ll. 324–325), coiled-coil domains are known to serve a variety of functions beyond filament formation. Therefore, it would be important to clarify why the authors favor the hypothesis of filament formation over alternative functional roles. Furthermore, to provide better context for this interpretation, a brief paragraph summarizing the biological roles and versatility of coiled-coil domains should be included in the introduction.

See lines 78-86 for an introductory paragraph introducing coiled-coil proteins including their diverse roles in bacterial biology.

This may be a matter of personal preference, but differential proteomics data is commonly presented as \log_2 fold changes (eg 10.1038/s41586-021-04003-2). More importantly, however, it is crucial to provide access to the complete proteomic dataset to ensure transparency and avoid the impression that only data supporting the authors' hypotheses has been selectively presented. Currently, it remains unclear whether other MGC-encoded proteins were also detected in the magnetosome-enriched or cell lysate fractions. Were these proteins identified but showed no significant difference between conditions, or were they not detected at all? I strongly recommend including a volcano plot, as shown in Ref. 10.1038/s41586-021-04003-2, to visualize not only the fold changes but also the statistical significance of the results, which is currently missing from the manuscript.

We have corrected the proteomics data to be presented with a \log_2 fold change (See Figures 3 as well as Supplemental Figures S3 and S11). We have also included volcano plots for entire proteomes for each proteomic experiment (Hydrogen vs Nitrogen, Early magnetosome fraction vs Early lysate, and Late magnetosome fraction vs. Late lysate – see Supplemental Figures S3 and S11). The MGC encoded proteins included in Supplemental Figure S3 were the only magnetosome proteins detected in our proteomic analyses. Mad25 was not included in Figure 3J because it was not more abundant at the magnetosomes in comparison to the cell lysate, but Mad25 was included in Supplemental Figure S3. An excel file with the complete data sets for each proteomic experiment is also provided. The excel file also includes tables of all magnetosome proteins detected.

The arrangement and organization of the figures could be improved, as they often do not follow the logical flow of the text (eg Fig. 2B is referenced after Figs. 2C and 2D). Although this is not a critical

issue, it makes the manuscript more difficult to follow, particularly when multiple images are referenced in quick succession (eg ll. 195–201).

We have attempted to fix these issues and can make more changes if needed.

Minor comments:

ll. 23-25: “These findings suggest that while biomineralization originates from a common ancestor, magnetosome chain formation has diverged evolutionarily among different MTB lineages.” – This is not a novel finding of the current study. Simple genome comparisons lead to this conclusion previously (e.g. 10.1093/nsr/nwac238). The sentence should therefore be **rephrased**.

Thank you for raising an important point regarding the novelty of our work and how it can be distinguished from other published studies. This is the first detailed genetic exploration of the function of key *mad* genes and reaches conclusions that are opposed to previous genomic and bioinformatic analyses. For instance, the excellent paper cited above assigns Mad10 as a biomineralization protein and Mad 23 as being involved in protein sorting. Our findings stand in stark contrast to these proposed functions. Additionally, the authors suggest that Mad25 and Mad26 are ATPases that could provide the energy needed to make more complicated chain structures (such as bundles). However, Mad25 and Mad26 are predicted to be entirely composed of coiled-coil domains and have no structural domains for ATP binding and hydrolysis. Finally, the cited paper proposes a conceptual framework for Mad28 and MamK function that is not supported by our data. We have added the following sentence to reflect these points (lines 83-85):

“Bioinformatic analyses of deep-branching MTB have proposed roles in biomineralization, protein sorting, and chain organization for coiled-coil-domain containing Mad proteins⁹. However, direct experimental evidence of their function has remained mysterious.”

Our data show that the steps and components used for chain formation in deep-branching MTB are distinct from those used by *Alphaproteobacteria*. This stands in contrast to biomineralization where many of the components are readily identifiable in all MTB as Mam protein homologs. Importantly, mutations in these *mam* genes also lead to similar phenotypes in RS-1 and AMB-1/MSR-1. Thus, we are suggesting that biomineralization has a common evolutionary origin whereas chain formation arose independently in deep-branching MTB as compared to the *Alphaproteobacteria*.

To address the reviewer’s concerns and enhance clarity, we have modified the sentence (lines 23-25) to read:

“These findings suggest that while biomineralization originates from a common ancestor, magnetosome chain organization has distinct evolutionarily origins among different MTB lineages.”

As stated above, we have also changed the discussion section to more directly highlight the differences between our findings and those of previous studies.

I. 43 “*Hydrpgendentota*” – correct to “*Hydrogenedentota*”

Fixed this see line 43.

ll. 54 “Among these, only five core genes (*mamA*, B, I, E, and Q) are shared across all MTB^{8,9}.” - Reference 9 shows more than 5 conserved genes. Moreover, while the absence of magnetosome

chains in Elusimicrobiota seem to support the absence of MamK, the genome sequences should be treated with caution due to incompleteness (ref 8).

We have altered the text to mention these discrepancies.

Lines 54-55 now read: “Depending on the accounting, only 5-9 core genes (*mamA*, *B*, *E*, *I*, *K*, *M*, *O*, *P* and *Q*) are shared across all MTB.”

Additionally, we addressed genome incompleteness in lines 388-390: “While genome incompleteness should be considered, the two *Elusimicrobiota* genomes show completeness levels of 75.84% and 94.38% in GTDB.”

II. 56 “mad (magnetosome deep branching)” genes - have been originally named ‘magnetosome associated Deltaproteobacteria’ in 10.1111/1462-2920.12128

Thank you for the correction. We have altered the text (lines 57-58).

I. 57 “man (magnetosome nitrospirota)” genes - have been originally named ‘magnetosome-associated *Nitrospirota*’ in 10.1186/s40168-024-01837-6

Thank you for the correction. We have altered the text (lines 58-59).

II. 67 “(Supplemental Figure S4E, Figure 2D-ii).” – Figure S4E does not exist!

Fixed to (Figure 2C-ii and 2G) on line 69.

I. 144 “revealed two mature magnetosome morphologies: curved and straight (Figure 3F ix and x).” – The magnetosomes at early biomineralization time points cannot be distinguished between curved and straight crystals. The panel is thus misleading to show early steps of curved magnetite crystals. Additionally, how can the authors be sure that not all crystals are curved when only looking at them in 2D-TEM?

This is a good point. While a sufficient number of crystals have not been 3D analyzed in RS-1 to definitively show that all crystals are straight in this organism, we do agree that with 2D TEM images also cannot determine 3D shapes. Our statement was to describe the crystals we were observing and a sentence has been added to acknowledge this uncertainty with 2D TEM images.

We have addressed this in lines 156-158.

II. 150 “In contrast, Alphaproteobacterial MTB such as AMB-1 and MSR-1, localize magnetosome membranes to the positive curvature before biomineralization begins^{2,21}. – Ref 2 is not appropriate for MSR-1. Additionally, in 10.1038/nature04382 it has been described that magnetosome formation in MSR-1 occurs throughout the cell!

This is an important point by the reviewer. In AMB-1, electron-cryotomography as well as fluorescence microscopy show that magnetosome membranes are organized as a chain prior to biomineralization. For MSR-1, the localization of magnetosomes prior to the initiation of biomineralization is less clear. The referenced manuscript does show seemingly random magnetosomes throughout the cell early after the addition of iron and invokes magnetic interactions as necessary for chain formation. However, subsequent studies

(<https://doi.org/10.1371/journal.pgen.1006101>) imply that magnetosome chains form at the positive cell curvature even in mutants with severe biomineralization defects. This manuscript also has an ECT image of a cell grown under aerobic (non-biomineralizing) conditions. Based on the image provided we cannot definitively state that the empty magnetosomes are at the positive cell curvature but they are close to the cell membrane. In RS-1, we have not yet been able to visualize empty magnetosome membranes but assume they are present based on the phenotypes of mutations in *mamB* and other transmembrane-domain-containing proteins. However, the data in the current manuscript clearly links progression in biomineralization to localization to the positive cell curvature. Given the different types of experiments performed in these three organisms it is perhaps premature to make the specific statement we presented in our first submission. We have thus removed it and present a more specific comparison between biomineralization and localization in the discussion section. See lines 338-342.

“Alphaproteobacterial” should not be italicized.

Corrected this.

II. 182 “In addition to the defects in crystal shape and size, *fmpA* and *fmpB* mutants exhibited abnormal crystal placement with most crystals not localized to the cell's positive curvature (Figure 4J).” - Ref. 12 and exemplary pictures in Fig 4A and B show mainly localization to the positive curvature and thus question the conclusion.

Supplemental Figure S2 has been added to show additional examples of *fmpA* and *fmpB* mutants and Figure 4A and B have been updated to complement the statistical figures showing that most crystals do not localize to the positive curvature.

II. 184 “The crystals of the *fmpA* mutant also fail to form chains, while only 5% of *fmpB* mutant cells show chain organization (Figure 4I).” - What is considered a chain? Isn't it obvious to have almost no chain formation with just 1 to 3 particles? In Ref. 12 chain formation is shown, indicating that chain formation is generally still possible.

For this manuscript, we defined chain formation as two or more crystals adjacent to one another. Supplemental Figure S2 B-i shows an example of a chain phenotype. The *fmpB* mutant makes many fewer crystals per cell than the WT but slightly more crystals per cell than the *fmpA* mutant. The image in Ref 12 is not generalizable since we had not carried out the extensive characterization of the mutants that are presented in this manuscript.

II. 187 “We propose that crystal nucleation occurs throughout the cell and crystals are later transported to the positive curvature post-maturation.” – What means post-maturation here? In TEM images one can also see immature crystals at the positive curvature. Please describe more specifically.

We have replaced “post-maturation” to “during the maturation process” to be more inclusive of our findings (line 202). It is meant as a general statement to encompass the totality of our results.

I. 193 “Mad10 is more abundant in early-stage magnetosome samples than in late stages ...” – Is this change significant. In Fig. S8A one can get the impression that there seem to be similar levels of Mad10 at magnetosomes.

The supplemental figure referenced (which has been moved to S3C) is comparing magnetosome fraction to cell lysate for early and late stages. To evaluate abundance of early vs. late proteins at the magnetosome, Figure 3J is the appropriate graph and that graph includes error bars for the standard deviation of biological replicates. The supplemental graph only gives information of how abundant a protein is at the magnetosome fraction in comparison to the cell lysate. Interestingly, Mad10 had the highest abundance at the magnetosome when compared to the cell lysate for any magnetosome protein identified in the proteomics.

I. 195 “Previous studies identified Mad10 as a magnetite-binding protein and hypothesized a role in magnetite nucleation or shape control.” – Similar assumptions have been made for MamJ (10.1038/nature04382). This might be a nice point to discuss.

As noted above, we have added a paragraph comparing Mad10 with MamJ in the discussion (lines 396-399). However, we do not know of experiments that definitively show magnetite binding for MamJ while this has been shown for Mad10.

II. 214 “In the $\Delta mad23$ mutant, single magnetosomes are dispersed, with the fewest crystals per subchain among mutants and WT.” – Here it should also be mentioned if the overall number of particles in the mutant is similar to WT or also reduced.

A sentence was added to describe this (lines 228-229).

II. 216 “The strain also has the lowest C_{Mag} of all chain organization mutants (Supplemental Figure 1B), suggesting Mad23 connects individual magnetosomes to form subchains.” – According to Fig. S1C $\Delta mad10$ has even lower C_{Mag}! How do you explain these discrepancies? How do the authors distinguish between the connection of magnetosomes into subchains versus the formation of a continuous chain in this case? In other words, the $\Delta mamK$ mutant also lacks subchains but still assembles a magnetosome chain, suggesting that subchain formation is not essential for chain formation. This implies that the role of Mad10 likely extends beyond merely promoting subchain formation.

Note: we have moved Supplemental Figure S1 to S4. When complementing mutants in RS-1, the kanamycin (in its sulfate salt form) used for plasmid selection leads to precipitation of iron and lower C_{mag}. This is most likely due to respiratory reduction of sulfate by RS-1 and reaction of the resulting sulfide with iron. All strains in S1B (now S4B) have no plasmids so they report a much more consistent C_{mag} value. All mutants in S1C (now S4C) are grown with a plasmid, either an empty plasmid as a control or a plasmid expressing the deleted gene leading to lower C_{mag} values. Additionally, we do not state that Mad10 promotes subchain formation. We propose that Mad10 coats the individual magnetosomes allowing for other chain organization proteins to interact with the magnetosomes. In our model we propose Mad23 interacts with Mad10, as verified with the bacterial two hybrid, and that interaction allows individual magnetosomes to start forming subchains. As for $\Delta mamK$, which shows one continuous chain, we propose that MamK is responsible for separating the subchains along the length of the cell and without MamK the subchains line up right next to or on top of other subchains. This hypothesis stems from the TEM analysis of chain organization and counting

the number of magnetosomes per cell. Data in Supplemental Figure S4 show a wide distribution of number of magnetosomes per cell leading to the hypothesis that MamK is important for spreading out the subchains so that daughter cells inherit roughly equal numbers of magnetosomes. In this way, one function of MamK seems to be conserved amongst MTB.

II. 228 “In AMB-1, for example, TEM shows subchains of magnetic particles, separated by empty magnetosome membranes all of which are organized into a continuous chain.” – Empty vesicles are usually not visible by conventional TEM.

We have removed the statement.

II. 249 “However, chains were no longer exclusively localized to the positive cell curvature and were found pole-to-pole at mid-cell, ...” – For better clarity, the manuscript should explicitly specify which cellular axis is meant when referring to the “mid-cell” position.

We have changed the statement to read: “pole-to-pole across the long axis at mid-cell” (line 266). There is also a supplemental figure describing these phenotypic categories (Supplemental figure S7 F-I).

II. 277 “Mad25 interacted with all tested proteins, while Mad20, 23, 26, and MamK interacted with three or more.” – Please describe in more detail.

Additional descriptions of the bacterial two hybrid was added (lines 294-301).

II. 287 “Our work suggests a new model of magnetosomes formation with the following steps (Figure 6):” – This sentence implies that the authors describe a new model for magnetosome formation that is valid for all MTB. However, the findings here are restricted to deep-branching MTB. The sentence should therefore be modified to clarify this fact.

This sentence was modified. See lines 306-307.

II. 289 “Biom mineralization initiates at randomly distributed membrane bounded nucleation sites that are likely membrane-enclosed.” – On which data or references are these assumptions based, particularly regarding the membrane-bound nucleation?

This is based on our previous work showing that mutations in genes encoding transmembrane-domain-containing Mam proteins (such as *mamB*) lead to severe defects in biomineralization including total absence of magnetite (Rahn-Lee et al. 2015).

II. 290 “Mutations in *fmpA* and *fmpB* impair crystal maturation, reduce crystal numbers, and disrupt localization to the positive curvature, linking biomineralization to chain formation.” – As mentioned above: Ref. 12 and exemplary pictures in Fig 4A and B show mainly localization to the positive curvature and sometimes chain formation. The conclusions should thus be discussed more critically.

We have changed the images chosen for *fmpA* and *fmpB* mutants in Figure 4A and B to better represent the statistics of cells with crystals that do not localize to the positive curvature. Additionally, another supplementary figure was added (Supplemental Figure S2) to show other images if *fmpA* and *fmpB* mutants with crystal localizations. While some crystals did localize to the positive curvature, most cells did not have positive curvature localization of crystals with more than 200 cells counted.

II. 296 “Bacterial two-hybrid results suggest an assembly pathway where Mad10 interacts with Mad23, which subsequently connects to Mad25.” - There is also direct Mad10-Mad25 interaction (Fig. S2A and D)!

The description of the model was changed to fix this issue. See lines 313-314.

II. 313 “AMB-1 and MSR-1 maintain a largely constant proteome^{2,33,34}. – Why mixing a review and primary literature here? References 33,34 are not proteomic studies and tested localization of few selected fluorescently labeled magnetosome proteins. The statement should therefore be expressed more carefully.

Thank you for this comment. We have removed the review. The main issue is that a similar study to ours is not available for AMB-1 and MSR-1. We are looking at the magnetosome proteome in RS-1 across a biomineralization time course. In AMB-1 and MSR-1 proteomic studies are conducted at a fixed timepoint when biomineralization has been completed. There is only one study we are aware of where magnetite particles in AMB-1 were separated based on their magnetic strength as a proxy for the magnetosome developmental stage. This study identified MamY as an “early” protein. Another study examined the magnetosome proteome of a MamE protease dead mutant (which has a severe biomineralization defect) and found only minimal differences as compared to the wildtype. References 33 and 34 are two studies that look at protein localization using fluorescently tagged magnetosome proteins in AMB-1 in non-biomineralizing and biomineralizing conditions. Their findings suggest that there are at least some proteins in AMB-1 are not present prior to the initiation of biomineralization meaning the magnetosome proteome may not be constant in the Alphaproteobacteria as well. We have amended the discussion (lines 343-358) to reflect these discrepancies.

II. 320 “Additionally, other than MamK, all known magnetosome chain formation factors of AMB-1 and MSR-1 (MamJ, MamY, LimJ, MamK-like, McaA, and McaB) are absent in RS-1.” - If even orthologs of MamJ and MamK are included among the factors mediating magnetosome chain formation, why are MamF-like proteins not mentioned?

This was an oversight on our part and we have now included the MamF-like proteins as well (line 386 and line 402).

II. 334 “This evolutionarily distinct solution may have arisen to combat the previously described conflict between magnetocrystalline orientation of tooth-shaped magnetosomes and their function as a navigational tool.” – Could this also simply be due to the increased magnetic forces imposed by the larger magnetite crystals?

As far as we can tell from the published literature, there is not a substantial difference in the magnetic forces of magnetosomes in RS-1 versus the model Alphaproteobacteria. However, the unusual cellular orientation of the easy vs hard magnetization axes has led to a model in which shape and chain formation compensate to allow for development of a dipole along the long axis of the cell.

II. 342 “... homologs for all five of the core mam genes selected: *mamK*, *mamA*, *mamB*, *mamM* and *mamQ*.” – The selected genes differ from those described in the introduction (*mamA*, *B*, *I*, *E*, and *Q*, l. 54). Explain why.

That was an error from an earlier version of the tree. Thank you for catching this. It has been addressed.

Figures 1 and S4: Different versions of the RS-1 MGC in different figures. Why?

Figure 1B: update nomenclature. Among others: LM-1 is known as *Candidatus Magnetocavibrio*

boulderlitore LM-1 10.1038/s41396-020-0647-x

Figure 1A was changed to have the same MGC as in Supplemental Figure S10 (previously S4). Additionally, we have update nomenclature in Figure 1B, thanks for bringing that to our attention.

Figure S3A: Which MamEO is shown here (C-term or N-term)? In Figure S4 MamEO-N-term is shown to have transmembrane domains.

In Supplemental Figure S1 (previously S3A) it is MamEO-Cterm– the figure has been corrected to show this. The proteins shown in that figure are proteins focused on in this study, so MamEO-Cterm was included because it was present in the proteomics.

Figure S7D: The WT is shown to have dispersed magnetosomes almost exclusively!?

Cells in TEM images often appear pretty dark which makes the identification of magnetosomes difficult.

Thank you for catching this! Figure S7D has been fixed. Also, the contrast of the TEM images have been adjusted for better visualization.

Reviewer #2 (Remarks to the Author):

Dr. Komeili's team has once again presented remarkable advances in the cell biology of magnetotactic bacteria. Russel et al. elegantly demonstrated the significance of Mad proteins in the formation and organization of magnetosomes in a magnetotactic bacterium from the Desulfobacterota phylum. They also proposed an interesting model for biomineralization in deep-branching magnetotactic bacteria (MTB). I was delighted to read the manuscript. However, I have a few minor comments:

We thank the reviewer for the kind words. We have addressed their comments below.

The authors should check the order of the supplemental figures and tables, along with the citations in the main text. For instance, I could not find citations for Supplemental Tables 5-8 within the main text.

These issues have been corrected.

Regarding scale bars, there seems to be inconsistency in their color, with some in black and others in white. The black scale bars are more visible (for example, in Fig. 2, Fig. 3F, 3H, Fig. 4A, and Fig. 4B). Additionally, the font size appears to differ in Fig. 2G i, ii, and iii compared to other scale bars in Fig. 2. Furthermore, scale bars are missing in Fig. 3A and 3B.

We changed all the scale bars to black and fixed any missing scale bars.

In the sentence, "Other sets of conserved genes, called mad (magnetosome deep branching) are found in all deep-branching MTB; man (magnetosome nitrospirota) genes are found in Nitrospirota, and mae (magnetosome elusimicrobiota) genes are found in Elusimicrobiota8," please include the reference for the mad and man genes. The abbreviation "mad" was initially used to refer to "magnetosome-associated Deltaproteobacteria" rather than "magnetosome deep branching." Are you proposing an updated abbreviation or do you have a reference for this change?

We have addressed this. See lines 57-58.

Regarding updates, it would be interesting to mention after the sentence about RS-1 strain classification (lines 64-65) that this bacterium was previously classified as Deltaproteobacteria because of the references.

We have addressed this see lines 65-67.

In the last sentence discussing the hydrogen effect on magnetosome synthesis (lines 117-118), it would be interesting to add the conclusion presented in the extended results section.

We have moved this conclusion into the main text (lines 126-129). However, to keep focus on the main results of the paper we have kept a more detailed paragraph in the extended data.

In the Electron Microscopy section, please consider adding a sentence indicating that statistics for magnetosome measurements are displayed in Tables 5-8.

We have added the statistical tables in the figure legend.

When scale bars are the same across a set of images (for example, Fig. 2E, Fig. 3F, Fig. 3H, Fig. 4G, Fig. 4H, and others), the authors could include just one scale bar and specify in the figure legend that this scale bar applies to the entire set of images. This approach may help clarify the figure by reducing visual elements. However, if the authors prefer to keep individual scale bars, note that scale bars are missing in Fig. 3F iv and viii.

We have fix missing scale bars and followed your advice in including only a single scale bare for the entire panel in Figure 3F. Thank you!

In Fig. 6, do the black circles in 1 and 2 represent a double-layered membrane? If so, the membrane should also have a label in the schematic representation.

Yes, they represent lipid bilayers and we have included membranes into the figure legend.

On line 765, "E. coli" should be italicized.

Corrected this. Thank you!

Lastly, in Supplemental S7, there is a missing space between the words "ChainPlacement."

Corrected this. Thank you!